# Genetic architecture of 11 organ traits derived from abdominal MRI using deep learning

Yi Liu[1], Nicolas Basty[2], Brandon Whitcher[2], Jimmy D Bell[2], Elena P Sorokin[1], Nick van Bruggen[1], E Louise Thomas[2†*], Madeleine Cule[1†*]

[1]Calico Life Sciences LLC, South San Francisco, United States; [2]Research Centre for Optimal Health, School of Life Sciences, University of Westminster, London, United Kingdom

**Abstract** Cardiometabolic diseases are an increasing global health burden. While socioeconomic, environmental, behavioural, and genetic risk factors have been identified, a better understanding of the underlying mechanisms is required to develop more effective interventions. Magnetic resonance imaging (MRI) has been used to assess organ health, but biobank-scale studies are still in their infancy. Using over 38,000 abdominal MRI scans in the UK Biobank, we used deep learning to quantify volume, fat, and iron in seven organs and tissues, and demonstrate that imaging-derived phenotypes reflect health status. We show that these traits have a substantial heritable component (8–44%) and identify 93 independent genome-wide significant associations, including four associations with liver traits that have not previously been reported. Our work demonstrates the tractability of deep learning to systematically quantify health parameters from high-throughput MRI across a range of organs and tissues, and use the largest-ever study of its kind to generate new insights into the genetic architecture of these traits.

*For correspondence:
l.thomas3@westminster.ac.uk
(ELT);
cule@calicolabs.com (MC)

†These authors contributed equally to this work

## Introduction

MRI is often regarded as the gold standard for the measurement of body composition in clinical research, with measurements of visceral adipose tissue (VAT), liver, and pancreatic fat content having an enormous impact on our understanding of conditions such as type-2 diabetes (T2D) and nonalcoholic fatty liver disease (NAFLD) (*Thomas et al., 2013*). In parallel to these developments, biobank-scale genome-wide association studies and epidemiological studies have elucidated the genetic basis of many complex traits, and shed light on their role in disease. The recent augmentation of the UK Biobank study with an imaging protocol has opened up many new avenues of research. In this work, we develop automated methods to quantify abdominal organ traits, characterise their genetic architecture, and explore their relationship to risk factors and disease outcomes.

The MRI protocol in the UKBB includes multiple tissues and organs with the potential for a wide variety of clinically relevant variables. However, genetic studies utilising the UKBB MRI-derived features have focused mainly on brain and cardiac traits (*Elliott et al., 2018*; *Miller et al., 2016*; *Pirruccello et al., 2020*), with some limited studies focussed on liver iron (n = 8,289) and MRI-based corrected T1 (n = 14,440) (*Parisinos et al., 2020*; *Wilman et al., 2019*). Thus, the full potential of the UKBB abdominal MRI data has not been realised, in part due to the lack of suitable automated methods to extract the variety and depth of relevant features from multiple organs in very large cohorts.

To address this issue, we trained models using deep learning on expert manual annotations, following preprocessing and quality control, to automatically segment key organs from the UKBB MRI data (*Table 1* and Materials and methods). Additionally, we quantified fat and iron content where

**Table 1.** Study population characteristics.
Age, BMI, and height rows give mean and SD for each population.

| | UK biobank cohort (at time of baseline visit) | Imaging cohort (at time of imaging visit) | GWAS cohort (White British Ancestry and passing QC) | | | |
| --- | --- | --- | --- | --- | --- | --- |
| | | | Organ volume (DIXON) | Pancreas volume | Pancreas fat and iron | Liver fat and iron |
| Number of participants | 502,520 | 38,881* | 32,860 | 31,758 | 25,617 | 32,858 |
| % Female | 54.4 | 51.8 | 51.5 | 51.4 | 51.2 | 51.5 |
| Age | 56.5 (8.1) | 63.7 (7.56) | 63.9 (7.52) | 63.8 (7.52) | 64.2 (7.48) | 63.9 (7.52) |
| BMI (kg/m$^2$) | 27.4 (4.8) | 26.5 (4.39) | 26.5 (4.37) | 26.5 (4.34) | 26.5 (4.31) | 26.5 (4.36) |
| Height (cm) | 168 (9.28) | 169 (9.3) | 169 (9.26) | 169 (9.25) | 169 (9.26) | 169 (9.26) |
| % White British Ancestry | 81.5 | 81.5 | 100 | 100 | 100 | 100 |

*Number of imaging participants gives the number with at least one abdominal IDP successfully extracted.

suitable acquisitions were available (*Figure 1—figure supplement 1a*, and Materials and methods). In total, we defined 11 Image Derived Phenotypes (IDPs): volume of the liver, pancreas, kidneys, spleen, lungs, VAT, and abdominal subcutaneous adipose tissue (ASAT), and fat and iron content of the liver and pancreas. By linking these traits to measures of risk factors, genetic variation, and disease outcomes, we are able to better characterise their role in disease risk.

## Results

*Table 1* characterises the study population compared to the entire imaging cohort. We were able to successfully extract IPDs from >99% of available scans for each modality (*Table 1* and *Supplementary file 1b*).

### Characterisation of IDPs in the UK biobank population

Previous studies have derived measures of VAT and ASAT, liver fat and iron in the UK Biobank from a subset of the scanned participants (*McKay et al., 2018*; *West et al., 2016*; *Wilman et al., 2017*). Our IDPs show a correlation of 0.87 (liver iron) to 1.0 (fat volume) (Materials and methods; *Figure 1—figure supplement 1*). The distribution of each organ-specific measure in the scanned population is summarised in *Figure 1E,F and G* and *Table 2*.

All IDPs, except liver fat, showed a statistically significant association with age after adjusting for imaging centre and date (*Figure 1B*), although the magnitudes of the changes are generally small (e.g. −8.8 ml or −0.03 s.d./year for liver volume, −27.7 ml or −0.0067 s.d./year for ASAT, and 24.3 ml or 0.011 s.d./year for VAT). Liver, pancreas, kidney, spleen, and ASAT volumes decreased, while VAT and lung volumes increased with age. Liver and pancreatic iron and pancreatic fat increase slightly with age. Several IDPs (volumes of liver, kidney, lung, and pancreas, as well as liver fat and iron) showed statistically significant evidence of heterogeneity in age-related changes between men and women. We found excess liver iron (>1.8 mg/g) in 3.22% of men and 1.75% of women.

To explore diurnal variation, we investigated correlation between the imaging timestamp and IDPs. We find a decrease in liver volume during the day, with volume at 12 noon being on average 112 ml smaller than volume at 8 am, and a return to almost the original volume by 8 pm. This has previously been suggested in small ultrasound studies (n = 8) which indicated that liver volume is at its smallest between 12 and 2 pm, attributed to changes in hydration and glycogen content (*Leung et al., 1986*). We also observe smaller, but still statistically significant, associations between time of day and liver and pancreas iron, as well as ASAT, VAT, kidney, and lung volume. Although these changes appear to be physiological in nature, we are currently unable to rule out other potential sources of confounding, however unlikely (for example, different groups of participants being more likely to attend the scanning appointment at different times of day).

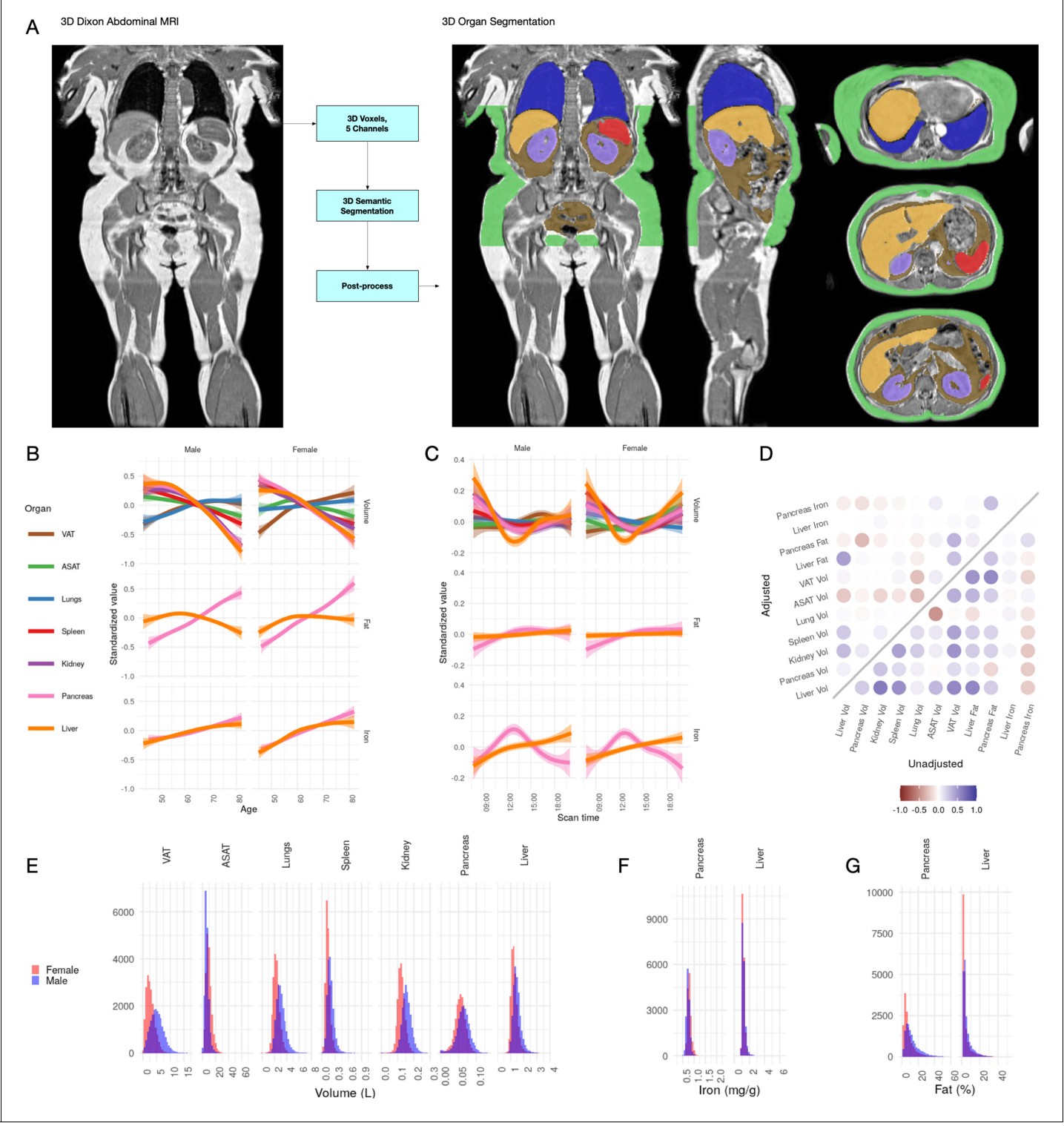

**Figure 1.** Visualisation of studied IDPs. (**A**) Example Dixon image before and after automated segmentation of ASAT, VAT, liver, lungs, left and right kidneys, and spleen. (**B**) Relationship between IDPs and age and sex within the UKBB. Each trait is standardised within sex, so that the y axis represents standard deviations, after adjustment for imaging centre and date. The trend is smoothed using a generalised additive model with smoothing splines for visualisation purposes. (**C**) Relationship between IDPs and scan time and sex within the UKBB. Each trait is standardised within sex, so that the y axis represents standard deviations, after adjustment for imaging centre and date. The trend is smoothed using a generalised additive model with smoothing splines for visualisation purposes. (**D**) Correlation between IDPs. Lower right triangle: Unadjusted correlation (except for imaging centre and

*Figure 1 continued on next page*

*Figure 1 continued*

date). Upper left triangle: Correlation after adjustment for age, sex, height, and BMI. (E-G) Histograms showing the distribution of the eleven IDPs in this study.

The online version of this article includes the following figure supplement(s) for figure 1:

**Figure supplement 1.** Correlation between multiple measurements of fat, iron and volume.

**Figure supplement 2.** IDPs plotted across imaging centre and across scan date.

## IDPs are associated with organ-specific disease outcomes

To assess which IDPs are associated with health-related outcomes, we defined a set of diseases based on inpatient hospital episode statistics (Materials and methods), and assessed the association between each IDP and disease diagnoses (*Figure 2* and *Supplementary file 1c*). Although we were not able to evaluate cause and effect, we found evidence that IDPs reflect organ function and health from the association with disease outcomes.

Liver volume was significantly associated with chronic liver disease and cirrhosis (p=4.5e-06, beta = 0.389) as well as T2D (p=1.3e-92, beta = 0.73) and hypertension (p=3.9e-17, beta = 0.18). Kidney volume was associated with chronic kidney disease (CKD) (p=8.0e-23, beta = −1.0). Interestingly, pancreas volume was associated more strongly with Type 1 diabetes (T1D) (p=4.9e-21, beta = −0.77, approximate 95% confidence interval [−0.93,–0.608]), than T2D (p=1.1e-17, beta = −0.27, approximate 95% confidence interval [−0.332,–0.208]). In contrast pancreatic fat showed a small association with T2D (beta = 0.181, p=1.16e-07) and not with T1D (p=0.241). Lung volume was most strongly associated with tobacco use (p=1.8e-46, beta = 0.50) and disorders relating to chronic airway obstruction (COPD) (p=3.6e-35, beta = 0.61), with larger lung volume corresponding to a greater likelihood of respiratory disease diagnosis. Spleen volume was associated with myeloproliferative disease (p=2.2e-33, beta = 0.74), especially chronic lymphocytic leukaemia (p=9.9e-24, beta = 0.78). Liver fat was associated with T2D (p=1.4e-34, beta = 0.29). Liver iron was associated with T2D (p=3.1e-19, beta = −0.43) and iron deficiency anaemia (p=5.3e-12, beta = −0.44) VAT was associated with a wide range of cardiometabolic outcomes including hypertension (p=1e-49, beta = 0.39), T2D (p=8.1e-44, beta = 0.69), and lipid metabolism disorders (p=1.9e-33, beta = 0.42), while ASAT was only associated with cholelithiasis and cholecystitis (p=1.3e-08, beta = 0.38). This association remained statistically significant, after adjusting for VAT, counter to reports that only VAT is predictive of gallstones (*Radmard et al., 2015*). Overall, this supports the key role of VAT and liver fat in the development of metabolic syndrome.

**Table 2.** Mean and standard deviations for 11 IDPs in our study, and number of independent GWAS associations found at study-wide significance (p<4.54e-9; see Materials and methods).

| Trait | Organ | Combined | Female | Male | # Study-wide significant GWAS hits |
|---|---|---|---|---|---|
| Volume (L) | VAT | 3.92 (2.3) | 2.78 (1.6) | 5.14 (2.3) | 3 |
| | ASAT | 8.16 (4.1) | 9.57 (4.3) | 6.64 (3.2) | 1 |
| | Lungs | 2.67 (0.73) | 2.32 (0.53) | 3.03 (0.75) | 5 |
| | Spleen | 0.17 (0.072) | 0.14 (0.054) | 0.2 (0.078) | 29 |
| | Kidney | 0.14 (0.03) | 0.12 (0.023) | 0.16 (0.028) | 9 |
| | Pancreas | 0.06 (0.018) | 0.06 (0.016) | 0.06 (0.019) | 11 |
| | Liver | 1.38 (0.3) | 1.28 (0.25) | 1.49 (0.3) | 11 |
| Fat (%) | Pancreas | 10.41 (7.9) | 8.34 (6.7) | 12.6 (8.5) | 8 |
| | Liver | 5.06 (5) | 4.43 (4.7) | 5.73 (5.2) | 11 |
| Iron (mg/g) | Pancreas | 0.77 (0.097) | 0.8 (0.1) | 0.75 (0.084) | 0 |
| | Liver | 1.22 (0.26) | 1.2 (0.24) | 1.24 (0.28) | 6* |

*Due to complex LD structure in this region, we were not able to finemap the HFE locus. We count two signals at this locus (rs1800562 and rs1799945).

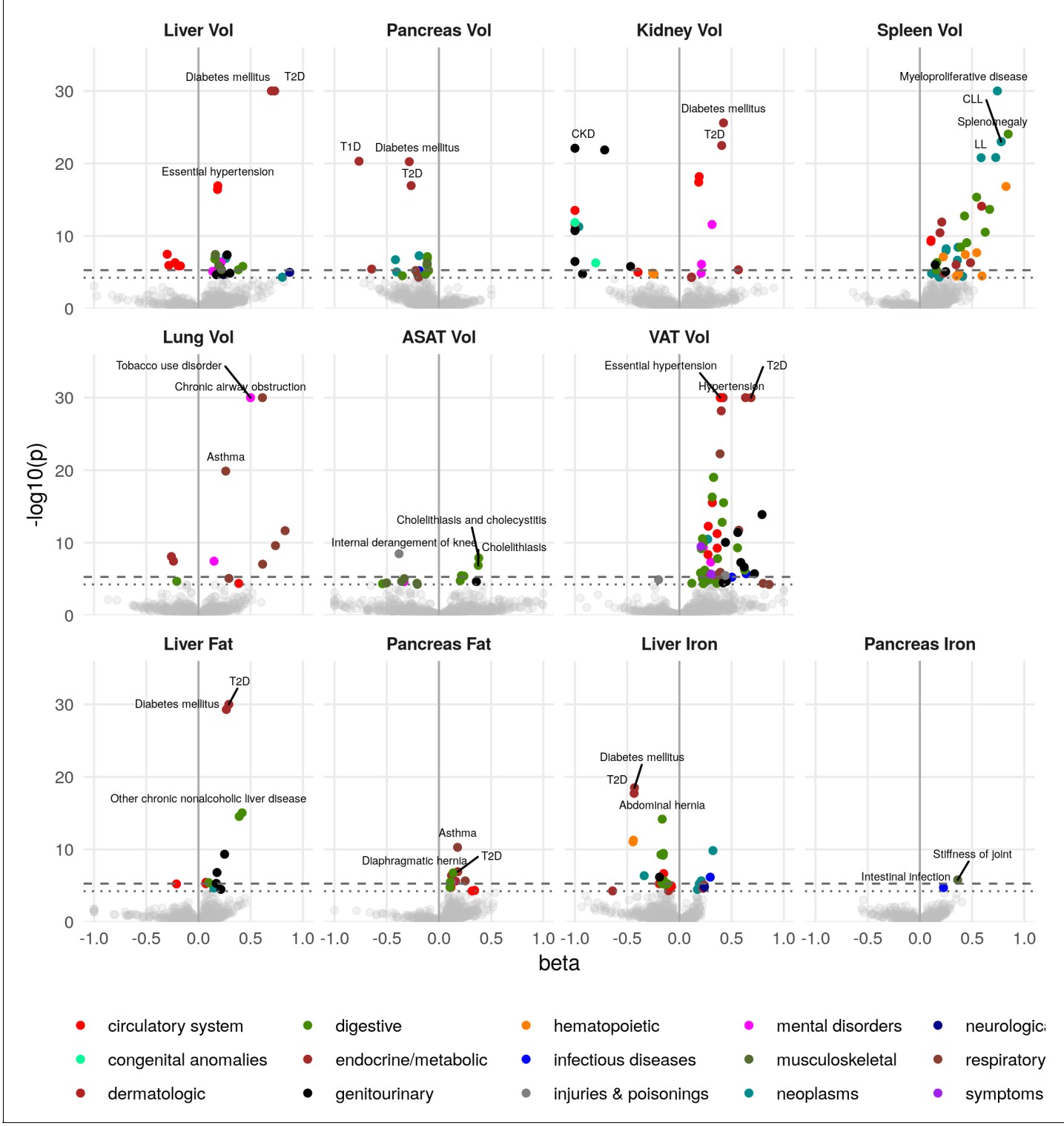

**Figure 2.** Disease phenome-wide association study across all IDPs and 754 disease codes (PheCodes). The x-axis gives the effect size per standard deviation, and the y-axis -log10(p-value). The top three associations for each phenotype are labelled. Horizontal lines at disease phenome-wide significance (dotted line, p=6.63e-05) and study-wide significance (dashed line, p=6.03e-06) after Bonferroni correction. Note that the PheCodes are not exclusive and have a hierarchical structure (for example, T1D and T2D are subtypes of Diabetes), so some diseases appear more than once in these plots. LL: Leukocytic leukaemia. CLL: Chronic leukocytic leakaemia. T1D: Type 1 diabetes. T2D: Type 2 diabetes. CKD: Chronic kidney disease. The online version of this article includes the following source data and figure supplement(s) for figure 2:

**Source data 1.** Source data for *Figure 2* and *Figure 2—figure supplement 1– 5*, phenome-wide association study across all IDPs.

*Figure 2 continued on next page*

*Figure 2 continued*

## IDPs are associated with organ-specific biomarkers, physiological measures, and behavioural traits

To further explore the extent to which our IDPs reflect organ health, we assessed correlation between the IDPs and 87 biomarkers from blood, serum, and urine, chosen to reflect a range of health conditions (Materials and methods, *Figure 2—figure supplement 1*, *Supplementary file 1d*). We also investigated associations between IDPs and 352 lifestyle and exposure factors, 844 self-reported medical history factors, 500 physical and anthropometric measures, and 769 self-reported diet and exercise measures (*Figure 2—figure supplements 3–5*, *Supplementary file 1d*).

Across multiple abdominal organs, we observed strong correlations between IDPs and biomarkers reflective of organ function. For example, liver volume was associated with triglycerides (p=1.19e-242, beta = 0.247) and sex hormone binding globulin (SHBG) (p=3.43e-210, beta = −0.216). Kidney volume was associated with serum cystatin C (p<1e-300, beta = −0.534), serum creatinine (p<1e-300, beta = −0.48), consistent with observations that smaller kidneys function less effectively (*Jovanović et al., 2013*). Pancreas volume was associated with glycated haemoglobin (HbA1c) (p=8.49e-28, beta = −0.0601), but the association with glucose was not statistically significant after Bonferroni correction (p=8.13e-05). Spleen volume was associated with multiple haematological measurements, including reticulocyte count (p<1e-300, beta = 0.25), mean sphered cell volume (p<1e-300, beta = −0.323), and platelet distribution width (p<1e-300, beta = 0.277).

Liver fat was associated with multiple liver function biomarkers including triglycerides (p=7.66e-219, beta-0.177), SHBG (p=4.75e-189, beta = −0.156) alanine aminotransferase (p<1e-300, beta = 0.226), and gamma glutamyltransferase (p=1.63e-194, beta = 0.162). Consistent with disease outcomes, which showed a correlation between hepatic iron, but not pancreatic iron, with iron deficiency anaemia, liver iron levels were correlated with measures of iron in the blood (e.g. mean corpuscular haemoglobin (MCH), p=1.71e-240, beta = 0.174), while pancreatic iron did not show any such association (MCH p=0.218).

Consistent with previous reports (*Harrison-Findik, 2007*), we found that liver iron was associated with lower alcohol consumption (p=3e-116, beta = −0.247) and higher intake of red meat (beef intake p=1.61e-61, beta = 0.168; lamb/mutton intake p=7.13e-56, beta = 0.165). Liver iron was also associated with suppressed T2* derived from neuroimaging in the same UKBB cohort (*Elliott et al., 2018*), particularly in the putamen (left: p=1.53e-68, beta = −0.138; right: p=1.01e-69, beta = −0.14). There were no such associations for pancreatic iron (left p=0.223; right p=0.194). Additionally, we found that liver fat was associated with lower birth weight (p=1.76e-30, beta = −0.0849) and comparative body size at age 10 (p=4.79e-76, beta = −0.22). Low birth weight has previously been associated with severity of pediatric non-alcoholic steatohepatitis (NASH) (*Bugianesi et al., 2017*), abnormal fat distribution (*Parkinson et al., 2020*), and liver fat levels in adults born prematurely (*Thomas et al., 2011*).

We found strong associations between increased lung volume and smoking status, tobacco smoking, COPD and lung disorders, wheeze, diagnosis of asthma and treatment for asthma, a decreased lung capacity as well as forced vital capacity (FVC) and forced expiratory volume in 1 s (FEV1)/FVC ratio (*Figure 2—figure supplement 5*). This is perhaps surprising in light of the age-related decreases in FEV1 and FVC; however, it has been shown that lung volume increases with both age and as a consequence of obstructive pulmonary diseases (*Lutfi, 2017*). Although lung volume estimated via MRI is not a widely used clinical measure, our data suggests it may be a biomarker of ageing-related respiratory complications.

## Genetic architecture of abdominal IDPs

To explore the genetic architecture of the IDPs, we performed a genome-wide association study (GWAS) for each IDP of 9 million single-nucleotide polymorphisms (SNPs) in the approximately 30,000 individuals of white British ancestry (*Bycroft et al., 2018*; Materials and methods). We verified that the test statistics showed no overall inflation compared to the expectation by examining the intercept of linkage disequilibrium (LD) score regression (LDSC) (*Bulik-Sullivan et al., 2015b*; *Supplementary file 1e*). Utilising a generalised linear mixed model framework and SKAT-O test implemented in SAIGE-GENE (*Zhou et al., 2020*), we performed gene-based exome-wide association studies in the 11,134 participants with IDP and exome sequencing data. Test statistics were well calibrated and we found no study-wide significant associations (*Figure 3—figure supplement 1*). The number of individuals included in the analysis for each IDP is given in *Table 1*, together with the number of study-wide significant independent signals for each IDP.

## Organ volume, fat, and iron are heritable

For each IDP, we estimated SNP-heritability using the BOLT-REML model (*Loh et al., 2015a*; Materials and methods). All IDPs showed a significant heritable component, indicating that genetic variation contributes substantially to the variation between individuals (*Figure 3A*). Heritability is largely unaffected by the inclusion of height and BMI as additional covariates, indicating that it is not a function of overall body size.

## Genetic correlation between abdominal IDPs

To understand the extent to which genetic variation explains the correlation between traits, we used bivariate LD score regression (*Bulik-Sullivan et al., 2015a*) to estimate the genetic correlation between all 55 IDP pairs, with and without including height and BMI as covariates (Materials and methods). After Bonferroni correction, we found a statistically significant non-zero genetic correlation between 22 of the 55 unadjusted IDP-pairs traits (*Figure 3B* and *Supplementary file 1f*), the strongest ($r_g$ = 0.782, p=4.60e-137) between ASAT and VAT. There was substantial genetic correlation between VAT and liver fat ($r_g$ = 0.58, p=3.7e-38) and between VAT and pancreas fat ($r_g$ = 0.569, p=2.79e-16). We found a negative genetic correlation between pancreas volume and fat ($r_g$ = −0.45, p=2.1e-06), and between pancreas volume and iron ($r_g$ = −0.5, p=5.2e-05).

## IDPs share a genetic basis with other physiological traits

To identify traits with a shared genetic basis, we estimated genetic correlation between IDPs and 282 complex traits with a heritable component (Materials and methods). A total of 650 IDP-trait pairs showed evidence of nonzero genetic correlation; 347 of these involved with measures of size or body composition (*Supplementary file 1g* and *Figure 3—figure supplement 2*). We found substantial genetic correlation between ASAT volume and other measures of body fat, such as whole-body fat mass ($r_g$ = 0.94, p=3.2e-143) and between VAT and conventional surrogate markers such as waist circumference ($r_g$ = 0.75, p=1.6e-109). The strongest genetic correlation with lung volume was with FVC ($r_g$ = 0.7, p=3.1e-71), with FEV and height also significant. We also found more modest genetic correlation between organ volumes and biochemical measures, such as liver fat and ALT ($r_g$ = 0.5, p=4.5e-23), kidney volume and serum creatinine ($r_g$ = −0.4, p=3.9e-22), and liver iron and erythrocyte distribution width ($r_g$ = −0.33, p=2.1e-14).

## Heritability is enriched in organ-specific cell types

In order to identify tissues or cell types contributing to the heritability of each trait, we used stratified LD score regression (*Finucane et al., 2015*) (Materials and methods). Liver fat showed evidence for enrichment in hepatocytes (p=4.20e-6) and liver tissue (p=2.2e-5), and pancreatic fat showed evidence for enrichment in pancreas tissue (smallest p=9.74e-5). Spleen volume showed enrichment in spleen cells (p=7.39e-10) and immune cell types including T cells, B cells, and natural killer cells, and neutrophils. VAT, ASAT, and lung volumes did not show evidence of significant heritability enrichment in any tissue or cell types (*Figure 3—figure supplements 3–5*).

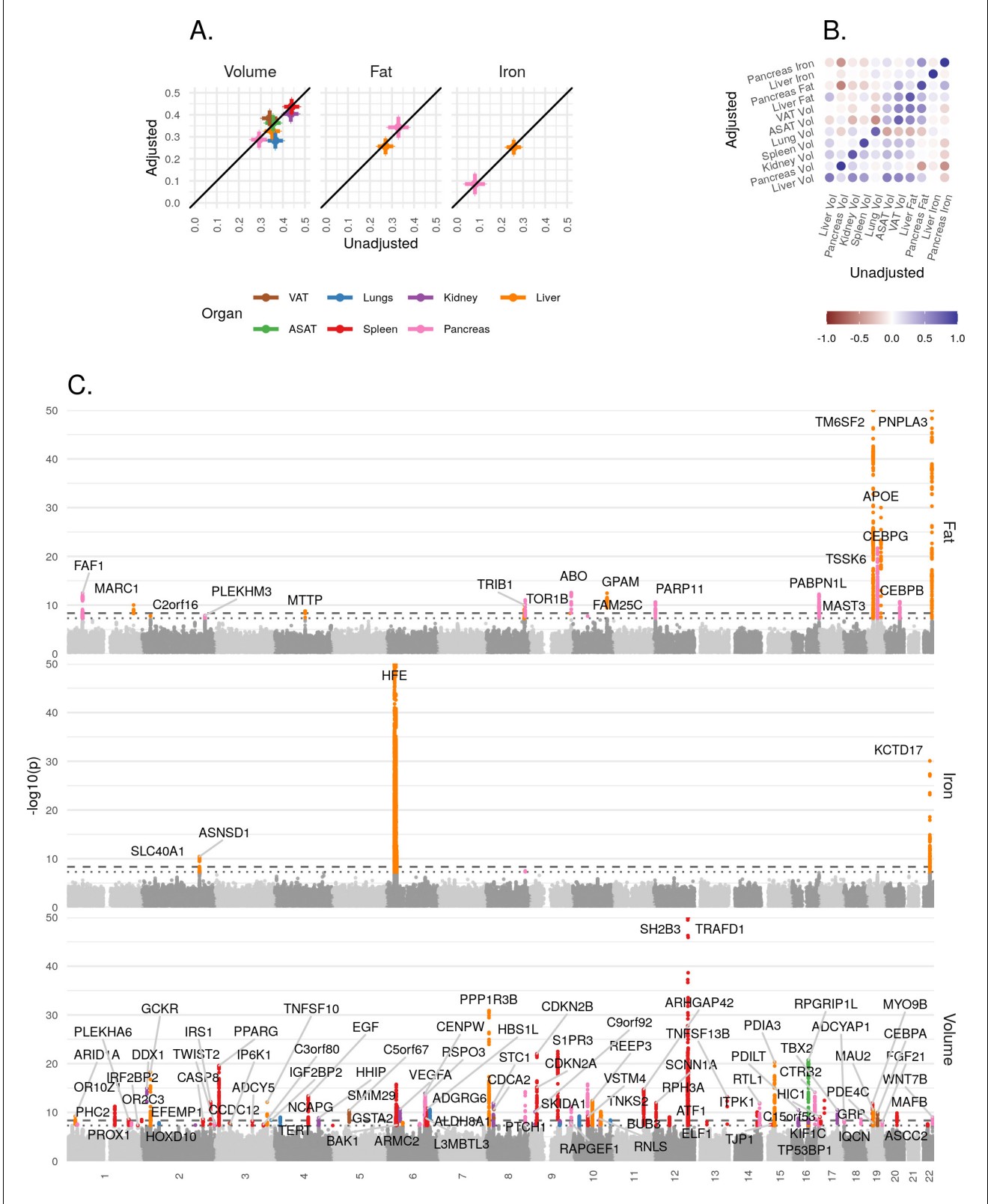

**Figure 3.** Genetic architecture of all IDPs. (**A**) Heritability (point estimate and 95% confidence interval) for each IDP estimated using the BOLT-REML model. Y-axis: Adjusted for height and BMI. X-axis: Not adjusted for height and BMI. The three panels show volumes, fat, and iron respectively. (**B**) Genetic correlation between IDPs estimated using bivariate LD score regression. The size of the points is given by -log10(p), where p is the p-value of the genetic correlation between the traits. Upper left triangle: Adjusted for height and BMI. Lower right triangle: Not adjusted for height and BMI. (**C**)

*Figure 3 continued on next page*

*Figure 3 continued*

Manhattan plots showing genome-wide signals for all IDPs for volume (top panel), fat (middle panel), and iron concentration (lower panel). Horizontal lines at 5e-8 (blue dashed line, genome-wide significant association for a single trait) and 4.5e-9 (red dashed line, study-wide significant association). P-values are capped at 10e-50 for ease of display. The genes with closest transcription start site are labelled.

The online version of this article includes the following figure supplement(s) for figure 3:

**Figure supplement 1.** Rare association studies in the subcohort with both exome sequence data and imaging-derived quantitative phenotypes.

**Figure supplement 2.** Genetic correlation between IDPs and complex traits.

**Figure supplement 3.** Heritability enrichment in tissues and cell types for annotations based on gene expression (see Materials and methods).

**Figure supplement 4.** Heritability enrichment in tissues and cell types for annotations based on chromatin accessibility (see Materials and methods).

**Figure supplement 5.** Heritability enrichment in tissues and cell types in immune cell types (see Materials and methods).

**Figure supplement 6.** QQ plots calculated based on a set approximately 500,000 LD-pruned, genotyped SNPs per trait.

## Genome-wide significant associations

For each locus containing at least one variant exceeding the study-wide significance threshold, we used GCTA COJO (*Yang et al., 2012*) to identify likely independent signals, and map likely causal variants (Materials and methods, *Supplementary file 1h*). To better understand the biology of each signal, we explored traits likely to share the same underlying signal (colocalised signals) among 973 traits and 356 diseases measured in UKBB (Materials and methods, *Supplementary file 1i*), and gene expression in 49 tissues (Materials and methods, *Supplementary file 1j*).

### Liver IDPs recapitulate known biology and point to new genes of interest

The strongest association with liver volume (lead SNP rs4240624, p=2.1e-34, beta = −0.15), lies on chromosome 8, 175 kb from the nearest protein-coding gene, *PPP1R3B*. *PPP1R3B* is expressed in liver and skeletal muscle, and promotes hepatic glycogen biosynthesis (*Mehta et al., 2017*). Although this variant has been associated with attenuated signal on hepatic computed tomography (*Stender et al., 2018*); in our study, it was not associated with liver fat (p=0.007) or iron (p=0.001).

We also detected an association between liver volume and a missense SNPs in *GCKR* (rs1260326, p=5.4e-19, beta = −0.061). This signal colocalised with T2D, hypercholesterolaemia and hyperlipidaemia, gout and gallstones, as well as other lipid and cardiovascular traits in the UKBB. This locus has previously been associated with NAFLD (*Kawaguchi et al., 2018*) as well as multiple metabolic traits including triglycerides, lipids, and C-reactive protein (*Wojcik et al., 2019*).

Of the eight study-wide independent signals associated with liver fat, three (rs58542926 in *TM6SF2* rs429358 in *APOE*; and rs738409 in *PNPLA3*) have previously been associated with NAFLD (*Kozlitina et al., 2014*; *Romeo et al., 2008*; *Speliotes et al., 2011*), and were also reported in a GWAS of liver fat in a subset of this cohort (*Parisinos et al., 2020*). The fourth SNP identified in that study, rs1260326 in *GCKR*, did not reach our stringent threshold of study-wide significance threshold (p=1.9e-8, beta = −0.044).

Two of the remaining five signals have previously been linked to liver disorders or lipid traits, although not specifically to liver fat. A signal near *TRIB1* (lead SNP rs112875651) colocalises with hyperlipidaemia and atherosclerosis and has been linked to lipid levels in previous studies, and SNPs in this gene have an established role in the development of NAFLD (*Liu et al., 2019*). A missense SNP in *TM6SF2* (lead SNP rs188247550) is also associated with hyperlipidaemia and has previously been linked to alcohol-induced cirrhosis (*Buch et al., 2015*).

Three further signals have not previously been associated with any liver traits, although some have been associated with other metabolic phenotypes. On chromosome 1, an SNP intronic to *MARC1* (lead SNP rs2642438) colocalises with cholesterol, LDL-cholesterol, and HDL-cholesterol levels, with the risk allele for higher fat associated with higher LDL-cholesterol. While this variant has not previously been associated with liver fat, missense and protein truncating variants in *MARC1* have been associated with protection from all-cause cirrhosis, and also associated with liver fat and circulating lipids (*Emdin et al., 2020*).

We found an association between intronic and *GPAM*, which encodes an enzyme responsible for catalysis in phospholipid biosynthesis (lead SNP rs11446981). This signal colocalises aspartate aminotransferase (AST), and HDL cholesterol levels in serum. *GPAM* knockout mice have reduced adiposity and its inhibition reduces food intake and increases insulin sensitivity in diet-induced obesity

(*Kuhajda et al., 2011*). Our data suggests that this enzyme may play a role in the liver fat accumulation in humans.

A region overlapping to *MTTP* with 67 variants in the 95% credible set was associated with liver fat. Candidate gene studies have linked missense mutations in *MTTP* to NAFLD (*Hsiao et al., 2015*). Rare nonsense mutations in this gene cause abetalipoproteinaemia, an inability to absorb and knockout studies in mice recapitulate this phenotype (*Partin et al., 1974*; *Raabe et al., 1998*). Inhibition of MTTP is a treatment for familial hypercholesterolaemia and is associated with increased liver fat (*Cuchel et al., 2007*).

We replicate previously reported associations with liver iron at *HFE* (rs1800562 and rs1799945) and *TMPRSS6* (*Wilman et al., 2019*), although we were unable to accurately finemap at the HLA locus. We found evidence for two independent additional signals on chromosome 2 between *ASND1* and *SLC40A1* (lead SNP rs7577758; conditional lead SNP rs115380467). *SLC40A1* encodes ferroportin, a protein essential for iron homeostasis (*Donovan et al., 2005*) that enables absorption of dietary iron into the bloodstream. Mutations in *SLC40A1* are associated with a form of haemochromatosis known as African Iron Overload (*Mayr et al., 2011*). This finding is consistent with a recent study which highlighted the role of hepcidin as a major regulator of hepatic iron storage (*Wilman et al., 2019*).

## Novel associations with pancreas IDPs

We identified 11 study-wide significant associations with pancreatic volume. None were coding or colocalised with the expression of protein-coding genes. Two signals (rs72802342, nearest gene *CTRB2*; rs744103, nearest gene *ABO*) colocalised with diabetic-related traits. This is consistent with our findings that T1D was associated with smaller pancreatic volume.

We identified seven study-wide significant independent associations with pancreatic fat, with little overlap with liver-specific fat loci. Surprisingly, we found little evidence that loci associated with pancreatic fat were associated with other metabolic diseases or traits, suggesting that it may have a more limited direct role in the development of T2D than previously suggested (*Taylor, 2008*).

The top association for pancreatic fat (lead SNP rs10422861) was intronic to *PEPD*, and colocalised with a signal for body and trunk fat percentage, leukocyte count, HDL-cholesterol, SHBG, total protein, and triglycerides. *PEPD* codes for prolidase, an enzyme that degrades iminopeptides in which a proline or hydroxyproline lies at the C-terminus, with a special role in collagen metabolism (*Kitchener and Grunden, 2012*). There was an association at the *ABO* locus (lead SNP rs8176685) for pancreatic fat; rs507666, which tags the A1 allele, lies in the 95% credible set at this locus. This signal colocalises with lipid and cardiovascular traits and outcomes, and is consistent with previous reports that blood group A is associated with lipid levels, cardiovascular outcomes (*Zhang et al., 2012*) and increased risk of pancreatic cancer (*Zhang et al., 2014*).

An association with pancreatic fat (lead SNP rs7405380) colocalises with the expression of *CBFA2T3* in the pancreas. rs7405380 lies in a promoter flanking region which is active in pancreatic tissue (ensemble regulatory region ENSR00000546057). *CBFA2T3* belongs to a family of ubiquitously expressed transcriptional repressors, highly expressed in the pancreas, about which little is known. A recent study identified Cbfa2t3 as a target of Hes1, which plays a critical role in regulating pancreatic development (*de Lichtenberg et al., 2018*). This SNP was not associated with any metabolic phenotypes.

We identified signals at a locus on chromosome 1 containing *FAF1* and *CDKN2C* (lead SNP rs775103516), and five other loci. In contrast to liver iron, where we identified strong signals at regions associated with ferroportin and hepcidin loci, we found no study-wide significant associations with pancreatic iron.

## Novel associations with other organ volume IDPs

A locus on chromosome 2 was associated with average kidney volume. This signal colocalises with biomarkers of kidney function (cystatin C, creatinine, urate, and urea) and a SNP in the 95% credible set, rs807624, has previously been reported as associated with Wilms tumor (*Turnbull et al., 2012*), a pediatric kidney cancer rarely seen in patients over the age of five. However, this association raises the possibility that this locus plays a broader role in kidney structure and function in an adult population and warrants further study.

We also found a significant association at the *PDILT/UMOD* locus (lead SNP rs77924615), that colocalises with hypertension, cystatin C, creatine, and kidney and urinary calculus in the UKBB. This locus has previously been associated with hypertension as well as estimated glomerular filtration rate (eGFR) and CKD (*Wuttke et al., 2019*) in other studies, supporting our finding that kidney volume reflects overall kidney function.

The trait with the most associations was the spleen, with 25 independent signals, of which 18 colocalised with at least one haematological measurement. We identified one association with ASAT volume (lead SNP rs1421085) at the well-known *FTO* locus which colocalised with many other body composition traits. The association with VAT volume at this SNP (p=3e-8, beta = 0.037) was not study-wide significant. We identified three additional signals associated with VAT volume. rs559407214 (nearest gene *CEBPA*) is independent of the nearby pancreatic fat signal. rs73221948 lies 150 kb from the nearest protein coding gene. This signal colocalises with triglyceride levels and HDL levels. This has previously been reported (*Richardson et al., 2020*), in addition to an association with BMI-adjusted waist-hip circumference (*Zhu et al., 2020*) Finally, rs72276239 which is also associated with trunk fat percentage, diabetes-related traits, cardiovascular problems, and lipids, and has previously been associated with waist-hip ratio (*Kichaev et al., 2019*).

## Discussion

We have developed a pipeline to systematically quantify organ and tissue parameters from MRI scans of over 38,000 participants in the UKBB imaging cohort, producing the largest sample size to date of abdominal imaging-derived phenotypes (IDPs). The training of our segmentation pipeline incorporated a broad range of data augmentation options, including smooth 3D geometric warps, to achieve better data efficiency. This enabled us to achieve good segmentation performance (Jaccard index >0.8) with a limited training dataset size of ~100 images. Since manual annotation of 3D images is a labor-intensive process, automating this process has removed a substantial barrier to large-scale studies of clinical images, and in turn facilitated new insights. The semantic segmentation models are robust to several sources of visual heterogeneity arising from deformable tissues and joints, and thus facilitate high-throughput analysis of MRI data.

The observed age-related decrease in organ volume (liver, pancreas, kidney, spleen) may reflect the predicted organ atrophy associated with ageing, likely underpinned by mechanism(s) similar to those reported for brain and skeletal muscle (*Mitchell et al., 2012*; *Svennerholm et al., 1997*). Individual organs exhibited distinct patterns of atrophy, with liver and pancreas exhibiting the largest reduction. The increase in VAT (but not ASAT) and lung volume with age may point at the overriding impact of environmental factors upon these tissues. Given that VAT and ASAT are exposed to similar exogenous factors, we hypothesise that the plasticity capacity of their adipocytes (hypertrophy and hyperplasia), and therefore tissue lipolysis and inflammation, ectopic fat deposition and insulin sensitivity, are differentially affected by the ageing process (*Mancuso and Bouchard, 2019*). Future studies which incorporate large-scale longitudinal imaging data will enable detailed interrogation of these changes between individuals.

The liver plays a pivotal role in the regulation of iron homeostasis, with iron excess to requirements stored in hepatocytes (*Anderson and Shah, 2013*). Epidemiologic studies utilising indirect methods based on serum markers (i.e. the ratio of serum transferrin receptor to serum ferritin) describe an age-related increase in total body iron, declining at a very late age (*Cook et al., 2003*). However, studies with direct measurements, although far more limited in scope and size, point towards a linear relationship with age (*Kühn et al., 2017*; *McKay et al., 2018*; *Nomura et al., 1988*; *Schwenzer et al., 2008*), similar to that observed in our study. The discrepancy between total and organ-specific changes with age may relate to the complex relationship between liver iron storage and circulating iron, which is known to be compromised by age related organ dysfunction and the inflammasome (*Anderson and Shah, 2013*). Similar patterns for pancreatic iron were observed (*Schwenzer et al., 2008*), again reflecting the overall iron homeostasis in the body.

Ectopic fat accumulation showed a more complex relationship with ageing. Although pancreatic fat increased with age for both men and women (*Schwenzer et al., 2008*), liver fat increased only up to approximately 60 years of age before plateauing in women and decreasing in men (*Kühn et al., 2017*; *Nomura et al., 1988*). Previous studies have suggested a linear relationship (*Thomas et al., 2012*; *Wilman et al., 2017*), but this may reflect the paucity of older participants

(>60 years) in those cohorts, thus lacking the power to detect the true effects of age on liver fat. Both liver fat and iron were associated with T2D, consistent with previous studies (*McKay et al., 2018*). No association was observed between pancreatic fat or iron content with either T1D or T2D, despite the observed association between pancreas volume and T1D. This is surprising given its proposed causal role assigned to this fat depot in T2D (*Taylor, 2013*). Interestingly, although both liver and pancreas volume decreased with age, pancreatic fat did not, in agreement with previous observations (*Majumder et al., 2017*). Additionally, there was considerably greater diurnal variation in liver volume compared with the pancreas. These observations add credence to the growing evidence of disparate mechanisms for the accumulation of fat in these organs (*Hellerstein, 1999*). Furthermore, given the observed diurnal variation in organ volume, fat and iron content, coupled to the known effects of feeding on the circadian clock on organ function (*Kalhan and Ghosh, 2015*), scheduling of MRI measurements of participants may be an important consideration in longitudinal studies.

Most organ volumes were associated with disease, highlighting the potential medical relevance of abdominal MRI-derived parameters. Associations with potential clinical relevance included kidney volume with chronic kidney disease (*Grantham et al., 2006*), and lung volumes with chronic obstructive pulmonary disease, bronchitis, and respiratory disease. Liver volume was associated with chronic liver disease (*Lin et al., 1998*) and cirrhosis (*Hagan et al., 2014*) as well as diabetes and hypertension. Although there is a strong correlation between liver volume and liver fat, liver volume is not generally measured in relation to metabolic disease. Whilst spleen volumes can be enlarged in response to a whole host of diseases such as infection, haematological, congestive, inflammatory, and neoplastic (*Pozo et al., 2009*), we found spleen volume to be most strongly associated with leukaemia. Although organ volume is not a widely-used measure for disease diagnosis, spleen volume is a useful metric for predicting outcome and response to treatment (*Shimomura et al., 2018*), and a robust automated measure of this IDP could be a powerful auxiliary clinical tool. Indeed, the associations with deep-learning derived organ and tissue parameters may become increasingly medically relevant in the future, as machine intelligence becomes more widely adopted as a component of clinical care.

The strong association between VAT and development of metabolic dysfunction is well established (*Lee et al., 2018*), and confirmed herein on a much larger cohort. No association between ASAT and disease, apart from incidence of gallstones, were observed. The overall role of subcutaneous fat in disease development is still debated. Viewed as benign or neutral in terms of risk of metabolic disease (*Kuk et al., 2006*), especially subcutaneous fat around the hips, ASAT does appear to be associated with components of the metabolic syndrome, though not after correcting for VAT or waist circumference (*Elffers et al., 2017*; *Irlbeck et al., 2010*). It has been suggested that subdivisions of ASAT may convey different risks, with superficial ASAT conferring little or no risk compared to deeper layers (*Kelley et al., 2000*). These conflicting results may reflect different approaches to ASAT and VAT measurement (MRI vs indirect assessment), size and make-up of study cohorts. Future studies within the UKBB and other biobanks will allow these relationships to be explored in more depth.

Through GWAS, we identify a substantial heritable component to organ volume, fat and iron content, both before and after adjusting for body size. We demonstrate heritability enrichment in relevant tissues and cell types (hepatocytes for liver fat, and pancreas for pancreatic fat), suggesting that there may be specific mechanisms underpinning organ morphology and function that warrant further investigation. For the traits that have been studied before in other cohorts, we replicate known associations such as the *PNPLA3, TM6SF2,* and *APOE* loci with liver fat, and of the *HFE* and *TMPRSS6* loci with liver iron. In addition, we identify several novel associations that may suggest mechanisms for further study, including an association between *GPAM* and liver fat, *PPP1R3B* and liver volume (but not liver fat), *CB2FAT3* and pancreatic fat, and *SLC40A1* and liver iron. Colocalisation analysis with gene expression in specific tissues implicated *CBFA2T3* in changes of pancreatic fat. We found little overlap between the significant loci for VAT, ASAT, liver fat, and pancreatic fat, highlighting the need to develop more refined definitions of adiposity to better understand the role it plays in disease risk. Our gene-based burden test for rare exome variants was limited by the smaller sample size available for this study. However, the substantial heritable component suggests that the planned studies involving up to 100,000 scanned individuals, including whole exome and whole genome

sequence data, will yield many further insights into the genetic basis of organ form, and its relationship to function.

This study has some limitations. Although recruitment into the UK Biobank study finished in 2010, scanning began in 2014. The median follow-up period from scanning is 2.5 years, limiting our power to evaluate the prognostic value of IDPs, or to evaluate whether they are a cause or consequence of the disease state. Since medical records will continue to be collected prospectively, we will be able to assess this more systematically in future studies. Our genetic studies were limited to participants of white British ancestry. While this did not greatly affect power due to the demographics of the imaging cohort, future imaging studies which incorporate greater diversity of ancestry and environmental exposure will facilitate fine-mapping as well as potentially elucidate new mechanisms (*Wojcik et al., 2019*). Additionally, we did not explore in detail the relationship between either ancestry or self-reported ethnicity, because of the limited sample size in the imaging cohort of non-White-British participants. Future studies with other cohorts could explore this question. Finally, while this study focussed on tractable measures derived from segmentation, we expect that future studies will allow us to define more sophisticated traits derived from organ segmentations and will give deeper insight into the relationship between organ form and function.

In conclusion, by systematically quantifying 11 IDPs covering several organs in the largest abdominal imaging cohort to date, we have associated organ parameters with environmental exposures, quantitative biomarkers, and clinical outcomes. In addition, we have characterised the genetic basis of these imaging-derived phenotypes to recapitulate previously identified associations with clinical endpoints, as well as uncover novel associations that may reflect new aspects of disease etiology or organ physiology. These findings could ultimately give insight into causes of complex disease, and potentially lead to new non-invasive diagnostic techniques. Moreover, the observations relating pancreatic volume to type-1 diabetes and liver volume with chronic liver disease along with gender differences, genetic susceptibility and volumetric changes related to diurnal variation will be important factors to consider for the growing field of personalised medicine. Deep-learning models trained on imaging data thus enhance our understanding of abdominal organ health and disease, and may guide strategies for personalised medicine or pave the way for new treatments in the future.

## Materials and methods

### Abdominal imaging data in UK biobank

All abdominal scans were performed using a Siemens Aera 1.5T scanner (Syngo MR D13) (Siemens, Erlangen, Germany). We analysed four distinct groups of acquisitions: (1) the Dixon protocol with six separate series covering 1.1 m of the participants (neck-to-knees), (2) a high-resolution T1-weighted (T1w) 3D acquisition of the pancreas volume, (3a) a single-slice multi-echo acquisition sequence for liver fat and iron, and (3b) a single-slice multi-echo acquisition sequence for pancreas fat and iron. Additional details of the MRI protocol may be found elsewhere (*Littlejohns et al., 2020*). The protocol covers the neck-to-knee region, including organs such as the lungs outside the abdominal cavity. For consistency with the UK Biobank terminology, we used the term 'abdominal' throughout the text.

The UK Biobank has approval from the North West Multi-centre Research Ethics Committee (MREC) to obtain and disseminate data and samples from the participants (http://www.ukbiobank.ac.uk/ethics/), and these ethical regulations cover the work in this study. Written informed consent was obtained from all participants.

### Image preprocessing

Analysis was performed on all available datasets as of December 2019, with 38,971 MRI datasets released by the UK Biobank, where a total of 100,000 datasets are the ultimate goal for the imaging sub-study. We focus here on four separate acquisitions, with one sequence being applied twice (once for the liver and once for the pancreas). The Dixon data were assembled into a single 3D volume for each participant using an automated fat-water swap detection and correction procedure. No additional preprocessing was necessary for the T1w 3D data for the pancreas. Proton density fat fraction (PDFF) and R2* were estimated from the single-slice multi-echo data for the liver and pancreas (*Bydder et al., 2020b*). The R2* values were converted into iron concentrations (*McKay et al.,*

*2018*; *Wood et al., 2005*). More details on the preprocessing steps may be found in the Supplementary Text.

## Manual annotation of abdominal structures for model training data

For each organ, we defined a standard operating procedure and provided training to a team of radiographers, utilising MITK, a free open-source software system for development of interactive medical image processing software (mitk.org). All annotations were visually inspected at multiple stages by experienced analysts before use in modelling.

## Segmentation of organs, for volume assessment, from Dixon data

We re-purposed an updated 3D iteration of the U-net architecture (*Ronneberger et al., 2015*) based on label-free segmentation from 3D microscopy (*Ounkomol et al., 2018*). Input voxels were encoded into five channels: fat, water, in-phase, out-of-phase, and body mask. The body mask indicated whether a given voxel was inside the body. To improve data efficiency, we pursued a multi-task approach (*Zhang and Yang, 2021*) and implemented aggressive data augmentation. We annotated multiple compartments and organs on the same individuals. Although not intrinsically novel, we are the first to scale this application to a very large UKBB imaging cohort. All weights are available to download (https://github.com/calico/ukbb-mri-sseg). This is the first time that segmentations for multiple major organs and compartments have been published on the UKBB dataset. Comparisons across datasets are also difficult because evaluation would be confounded by the specifics of how individuals are chosen, the conventions of annotation, and specifics of data acquisition or processing.

### Abdominal subcutaneous adipose tissue (ASAT) and visceral adipose tissue (VAT)

Two structures, the 'body cavity' and 'abdominal cavity', were segmented using neural-network based methods from the Dixon segmentation to estimate ASAT and VAT. For estimation of VAT, the abdominal cavity was used to isolate only tissue in the abdomen and pelvis. The fat channel was thresholded, small holes filled, and segmentations of abdominal organs (e.g. liver, spleen, kidneys) were removed to produce the final mask of VAT. For ASAT estimation, the body cavity was used to exclude all tissue internal to the body. A bounding box was computed based on the abdominal cavity, where the upper and lower bounds in the superior-inferior (z) direction were used to define the limits of the ASAT compartment.

### Segmentation of the liver, for fat and iron content assessment, from single-slice data

To automatically segment livers on 2D liver acquisitions, we trained one 2D U-net model with standard data augmentations for IDEAL, and another model for GRE. During inference, we ensured high specificity, at the cost of recall, by ablating the foreground mask by 25%. We made this trade-off because it is critical to include only liver tissue in the downstream analysis. In addition we removed voxels with R2* values outside the physiological range [18.78, 68.9] (*McKay et al., 2018*). Final values were not sensitive to this filter.

### Pancreas segmentation from T1w MRI (volume) and extraction (fat and iron content assessment), from single-slice data

We performed pancreas 3D segmentation on the high-resolution T1w 3D acquisition based on a recent iteration of the U-net architecture used in label-free microscopy (*Ounkomol et al., 2018*), using 123 manual annotations. Segmentation was not performed using the Dixon data since the pancreas has a complex morphology and benefited from improved contrast and resolution. The segmented volume was resampled to extract an equivalent 2D mask for the single-slice data (*Basty et al., 2020*).

## Statistical analysis of IDPs

All statistical analyses were performed using R version 3.6.0.

## Comparison with previous studies

We compared the values extracted in our study with those from previous studies, available from the following UK Biobank fields:

- VAT (Field 22407) and ASAT (Field 22408) (*West et al., 2016*)
- Liver fat (22400) and liver iron (22402) (*Wilman et al., 2017*)

## Relationship between age, scan time, and IDPs

For fitting linear models, we used the R function 'lm'. For fitting smoothing splines, we used the 'splines' package. To determine whether a coefficient was statistically significant in a set of models, we adjusted the p-values for each coefficient using Bonferroni correction. We compared models with and without scan time using ANOVA.

We looked for systematic differences between scanning centre, and trends by scan date (*Figure 1—figure supplement 2*). Because there were some minor differences unlikely to be of biological interest, we included scanning centre and scan date as covariates in all subsequent analyses.

## Disease phenome defined from hospital records

We used the R package PheWAS (*Carroll et al., 2014*) to combine ICD10 codes (Field 41270) into distinct diseases or traits (PheCodes). The raw ICD10 codes were grouped into 1283 PheCodes; of these, 754 PheCodes had at least 20 cases for all IDPs dataset allowing for a meaningful regression model. For each IDP-PheCode pair, we performed a logistic regression adjusted for age, sex, height, and BMI, and imaging centre and imaging date, scan time, and self-reported ethnicity.

We defined two Bonferroni-adjusted p-values: a single-trait value of 6.63e-5, and a study-wide value of 6.03e-6. As many of the diagnoses are correlated, we expect this threshold to be conservative.

## Other traits

We used the R package PHESANT (*Millard et al., 2018*) to generate an initial list of variables derived from raw data. We manually curated this list to remove variables related to procedural metrics (e.g. measurement date, time and duration; sample volume and quality), duplicates (e.g. data collected separately on a small number of participants during the pilot phase), and raw measures (e.g. individual components of the fluid intelligence score). This resulted in a total of 1824 traits. For each trait, we performed a regression (linear regression for quantitative traits, and logistic regression for binary traits) on the abdominal IDP, including imaging centre, imaging date, scan time, age, sex, BMI, and height, and self-reported ethnicity as covariates.

We defined two Bonferroni-adjusted p-values: a single-trait value of 2.75e-5, and a study-wide value of 2.49e-6. As many traits are correlated, we expect this threshold to be conservative.

## Genetics

We follow the methods described in a previous study (*Sethi et al., 2020*).

### Genome-wide association study

We used the UKBB imputed genotypes version 3 (*Bycroft et al., 2018*), excluding single nucleotide polymorphisms (SNPs) with minor allele frequency <1% and imputation quality <0.9. We included only participants who self-reported their ancestry as 'White British' and who clustered with this group in a principal components analysis (*Bycroft et al., 2018*). We excluded participants exhibiting sex chromosome aneuploidy, with a discrepancy between genetic and self-reported sex, heterozygosity and missingness outliers, and genotype call rate outliers (*Bycroft et al., 2018*). We used BOLT-LMM version 2.3.2 (*Loh et al., 2015b*) to conduct the genetic association study. To calculate the genotype-relatedness matrix, we followed the recommendation of the BOLT-LMM authors and used an LD-pruned (r2 <0.8) set of 574,316 SNPs extracted from the genotyped SNPs and a leave-one-chromosome-out (LOCO) approach to test association with each SNP. We included age at imaging visit, age squared, sex, imaging centre, scan date, scan time, and genotyping batch as fixed-effect covariates, and genetic relatedness derived from genotyped SNPs as a random effect to control for population structure and relatedness. The genomic control parameter, computed from an

LD-pruned set of genotyped SNPs ranged from 1.02 to 1.09 across eleven IDPs (*Supplementary file 1k* and *Figure 3—figure supplement 6*). We verified that the test statistics showed no overall inflation compared to the expectation by examining the intercept of linkage disequilibrium (LD) score regression (LDSC) (*Bulik-Sullivan et al., 2015b*; *Supplementary file 1e*), suggesting that the slightly inflated GC parameter is likely due to the polygenicity of these traits, rather than residual confounding. In addition to the commonly-used genome-wide significance threshold of p=5e-8, we defined an additional study-wide significance threshold using Bonferroni correction for the number of traits, p=5e-8/11 = 4.5e-9. For this analysis and all other analyses using LDSC, we followed the recommendation of the developers and (i) removed variants with imputation quality (info) <0.9 because the info value is correlated with the LD score and could introduce bias, (ii) excluded the major histocompatibility complex (MHC) region due to the complexity of LD structure at this locus (GRCh37::6:28,477,797–33,448,354; see https://www.ncbi.nlm.nih.gov/grc/human/regions/MHC), and (ii) restricted to HapMap3 SNPs (*Altshuler et al., 2010*).

For each IDP, we performed a secondary analysis with height and BMI as additional covariates.

## Exome-wide association study

Exome sequencing variant calls from the raw FE variant calling pipeline (*Regier et al., 2018*) were downloaded from the UK Biobank website (http://biobank.ctsu.ox.ac.uk/crystal/field.cgi?id=23160). QC was performed in PLINK v.1.90 using the following criteria: removal of samples with discordant sex (no self-reported sex provided, ambiguous genetic sex, or discordance between genetic and self-reported sex), sample-level missingness <0.02, European genetic ancestry as defined by the UK Biobank (*Bycroft et al., 2018*). Variant annotation was performed using VEP v100, filtered for rare (MAF <0.01) putative loss-of-function variants including predicted high-confidence loss-of-function variants, predicted using the LOFTEE plugin (*Karczewski et al., 2020*). A total of 11,134 samples and 11,939 genes were analysed in a generalised linear mixed model as implemented in SAIGE-GENE (*Zhou et al., 2020*). A filtering step of at least five loss-of-function carriers per gene was applied, resulting in 6745 genes. A kinship matrix was built in SAIGE off of a filtered set of array-genotyped variants ($r^2$ <0.2, MAF $\geq$ 0.05, autosomal SNPs, exclusion of regions of long-range LD, HWE p>1e-10 in European population). Outcome variables were inverse normal transformed and regressed on gene carrier status, adjusted for genetic sex, age, $age^2$, the first 10 principal components of genetic ancestry, scaled scan date, scaled scan time, and study centre as fixed effects and genetic relatedness as a random effects term.

## Heritability estimation and enrichment

We estimated the heritability of each trait using restricted maximum likelihood as implemented in BOLT version 2.3.2 (*Loh, 2018*).

To identify relevant tissues and cell types contributing to the heritability of IDPs, we used stratified LD score regression (*Finucane et al., 2018*) to examine enrichment in regions of the genome containing genes specific to particular tissues or cell types. We used three types of annotations to define: (i) regions near genes specifically expressed in a particular tissue/cell type, (i) regions near chromatin marks from cell lines and tissue biopsies of specific cell types, and (iii) genomic regions near genes specific to cells from immune genes. For functional categories, we used the baseline v2.2 annotations provided by the developers (https://data.broadinstitute.org/alkesgroup/LDSCORE). Following the original developers of this method (*Finucane et al., 2018*), we calculated tissue-specific enrichments using a model that includes the full baseline annotations as well as annotations derived from (i) chromatin information from the NIH Roadmap Epigenomic (*Kundaje et al., 2015*) and ENCODE (*ENCODE Project Consortium, 2012*) projects (including the EN-TEx data subset of ENCODE which matches many of the GTEx tissues, but from different donors), (ii) tissue/cell-type-specific expression markers from GTEx v6p (*GTEx Consortium et al., 2017*) and other datasets (*Fehrmann et al., 2015*; *Pers et al., 2015*), and (iii) immune cell type expression markers from the ImmGen Consortium (*Heng et al., 2008*). For each annotation set, we controlled for the number of tests using the Storey and Tibshirani procedure (*Storey and Tibshirani, 2003*). Although heritability is non-negative, the unbiased LDSC heritability estimate is unbounded; thus, it is possible for the estimated heritability, and therefore enrichment, to be negative (e.g. if the true heritability is near zero and/or the sampling error is large due to small sample sizes).

To enable visualisation, we grouped tissue/cell types into systems (e.g. 'blood or immune', 'central nervous system') as used in *Finucane et al., 2018*.

## Genetic correlation

We computed genetic correlation between traits using bivariate LDSC (*Bulik-Sullivan et al., 2015a*).

## Statistical fine-mapping

We performed approximate conditional analysis using genome-wide complex trait analysis (GCTA) (*Yang et al., 2012*), considering all variants that passed quality control measures and were within 500 kb of a locus index variant. As a reference panel for LD calculations, we used genotypes from 5,000 UKBB participants (*Bycroft et al., 2018*) that were randomly selected after filtering for unrelated participants of white British ancestry. We excluded the major histocompatibility complex (MHC) region due to the complexity of LD structure at this locus (GRCh37::6:28,477,797–33,448,354; see https://www.ncbi.nlm.nih.gov/grc/human/regions/MHC). For each locus, we considered variants with genome-wide evidence of association (Pjoint $<10^{-8}$) to be conditionally independent. We annotated each independent signal with the nearest known protein-coding gene using the OpenTargets genetics resource (May 2019 version).

## Construction of genetic credible sets

For each distinct signal, we calculated credible sets (*Maller et al., 2012*) with 95% probability of containing at least one variant with a true effect size not equal to zero. We first computed the natural log approximate Bayes factor (*Wakefield, 2007*) $\Lambda_j$, for the j-th variant within the fine-mapping region:

$$\Lambda_j = \ln\left(\sqrt{\frac{V_j}{V_{j+\omega}}}\right)\frac{\omega\beta^2}{2V_j(V_j+\omega)}$$

where $\beta_j$ and $V_j$ denote the estimated allelic effect (log odds ratio for case control studies) and corresponding variance. The parameter $\omega$ denotes the prior variance in allelic effects and is set to $(0.2)^2$ for case control studies (*Wakefield, 2007*) and $(0.15\sigma)^2$ for quantitative traits (*Giambartolomei et al., 2014*), where $\sigma$ is the standard deviation of the phenotype estimated using the variance of coefficients (Var($\beta_j$)), minor allele frequency ($f_j$), and sample size ($n_j$; see the sdY.est function from the coloc R package):

$$2n_jf_j(1-f_j)\sim\sigma^2\frac{1}{Var(\beta_j)}-1$$

Here, $\sigma^2$ is the coefficient of the regression, estimating $\sigma$ such that $\sigma = \sqrt{\sigma^2}$.

We calculated the posterior probability, $\pi_j$, that the j th variant is driving the association, given l variants in the region, by:

$$\pi_j = \frac{(1-\gamma)\Lambda_j}{l\sum_{k=0}^{l}\Lambda_k}$$

where $\gamma$ denotes the prior probability for no association at this locus and k indexes the variants in the region (with k = 0 allowing for the possibility of no association in the region). We set $\gamma = 0.05$ to control for the expected false discovery rate of 5%, since we used a threshold of P marginal $<5\times10^{-8}$ to identify loci for fine-mapping. To construct the credible set, we (i) sorted variants by increasing Bayes factors (natural log scale), (ii) included variants until the cumulative sum of the posterior probabilities was $\geq 1-c$, where c corresponds to the credible set cutoff of 0.95.

## Colocalisation of independent signals

To identify other traits potentially sharing the same underlying causal variant, we downloaded a catalog of summary statistics using the UK Biobank cohort from http://www.nealelab.is/uk-biobank (Version 2). For disease phenotypes, we additionally downloaded summary statistics computed using SAIGE (*Zhou et al., 2018*) from https://www.leelabsg.org/resources. After de-duplication, removal

of biologically uninformative traits, and removal of traits with no genome-wide significant associations, we considered a total of 974 complex traits and, and 356 disease phenotypes. To identify potentially causal genes at each locus, additionally explored expression QTL data from GTEx (version 7, dbGaP accession number dbGaP accession number phs000424.v7.p2) to seek evidence for colocalisation with expression in one of 49 tissues.

We performed colocalisation analysis using the coloc R package (*Giambartolomei et al., 2014*) using default priors and all variants within 500 kb of the index variant of each signal. Following previous studies (*Guo et al., 2015*), we considered two genetic signals to have strong evidence of colocalisation if PP3+PP4$\geq$0.99 and PP4/PP3 $\geq$5.

## Identifying other associations with our lead signals

In addition to the colocalisation analysis with UK Biobank traits, order to identify GWAS signals tagged by any of our associations from previous studies (not including the UK Biobank traits described above), we queried the Open Targets Genetics Resource (*Carvalho-Silva et al., 2019*), version 190505. We identified for studies where our lead variant was in LD (r > 0.7) with the lead SNP of a published study. We also searched for our lead SNPs in the NHGRI-EBI GWAS catalog (*Buniello et al., 2019*) in October 2020.

## Code availability

MATLAB code to estimate the PDFF is available from Dr Mark Bydder at https://github.com/marc-sous/pdff (*Bydder, 2020a*).

Code to preprocess the imaging data is available from https://github.com/recoh/pipeline (*Whitcher and Basty, 2021*; copy archived at swh:1:rev:13dc77941cb2919417108637ea-de6c8448374229). Fitted models and code to apply the models is available from https://github.com/calico/ukbb-mri-sseg/ (*Liu, 2021*; copy archived at swh:1:rev:4acdad6bf5e6cd08436d91ac6d4a494cf1365d98).

# Acknowledgements

We thank Adam Baker, Garret Fitzgerald, Frank Li, Anil Raj, and Amoolya Singh for input on the manuscript, and Leland Taylor for writing the genetic analysis pipeline used in this manuscript. We thank Stefan Stender for feedback on a draft. We thank the Edward Janus (Reviewing Editor), Matthias Barton (Senior Editor), Constantinos Parisinos (Reviewer) and one anonymous reviewer for their feedback on the manuscript. The following individuals involved in review of your submission have agreed to reveal their identity: Constantinos Parisinos This study was carried out using UK Biobank Application number 44584, and we thank the participants in the UK Biobank imaging study. This study was funded by Calico Life Sciences LLC.

# Additional information

### Competing interests

Yi Liu, Nick van Bruggen, Madeleine Cule: Employee, Calico Life Sciences LLC. This work was funded by Calico Life Sciences LLC. Elena P Sorokin: Employee, Calico Life Sciences LLC.This work was funded by Calico Life Sciences LLC. The other authors declare that no competing interests exist.

### Funding

No external funding was received for this work.

### Author contributions

Yi Liu, Software, Formal analysis, Investigation, Visualization, Methodology, Writing - original draft; Nicolas Basty, Brandon Whitcher, Data curation, Software, Formal analysis, Investigation, Visualization, Methodology, Writing - original draft, Writing - review and editing; Jimmy D Bell, Conceptualization, Resources, Supervision, Funding acquisition, Investigation, Project administration, Writing - review and editing; Elena P Sorokin, Software, Formal analysis, Visualization, Methodology, Writing -

original draft, Writing - review and editing; Nick van Bruggen, Conceptualization, Resources, Supervision, Writing - review and editing; E Louise Thomas, Conceptualization, Data curation, Supervision, Investigation, Methodology, Writing - original draft, Project administration, Writing - review and editing; Madeleine Cule, Conceptualization, Data curation, Software, Formal analysis, Validation, Investigation, Visualization, Methodology, Writing - original draft, Project administration, Writing - review and editing

## Author ORCIDs

Yi Liu (iD) https://orcid.org/0000-0003-2745-6940
Nicolas Basty (iD) http://orcid.org/0000-0002-1330-0913
Brandon Whitcher (iD) https://orcid.org/0000-0002-6452-2399
Jimmy D Bell (iD) https://orcid.org/0000-0003-3804-1281
Elena P Sorokin (iD) http://orcid.org/0000-0001-8957-8869
E Louise Thomas (iD) https://orcid.org/0000-0003-4235-4694
Madeleine Cule (iD) https://orcid.org/0000-0002-7400-5643

## Decision letter and Author response
Decision letter https://doi.org/10.7554/eLife.65554.sa1
Author response https://doi.org/10.7554/eLife.65554.sa2

# Additional files

## Supplementary files

• Supplementary file 1. (a) Segmentation performance metrics. (b) Numbers of participants at each stage of the processing pipeline for different scan and data types. (c) Significant PheWAS associations. Only associations which are statistically significant after correction for multiple testing are shown. (d) Significant PHESANT associations. Only associations which are statistically significant after correction for multiple testing are shown. (e) LDSC intercept. (f) Genetic correlations between abdominal IDPs. (g) Genetic correlation between abdominal IDPs and other heritable complex traits. Only associations which are statistically significant after correction for multiple testing are shown. (h) Genome-wide significant lead SNPs. Columns are as follows • trait: One of: volume, fat or iron • organ: Organ • var_index Index variant (in the format chr:pos:ref:alt:build) (All index variants are listed in GRCh37 coordinates) • rs_id: dbSNP ID • var_conditional: If a conditional signal, variants conditioned on, in the same format as var_index • pv P-value • pp: Probability that the lead SNP is the causal variant • beta: Effect size (in standard deviations) • closest_gene: Closest protein-coding gene • closest_gene_dist: Distance to TSS of closest gene (i) Significant colocalisation with complex trait GWAS signals. (j) Significant colocalisation with gene expression. (k) Genomic control parameter for each trait computed using BOLT-LMM on an LD-pruned set of genotyped SNPs.

• Transparent reporting form

## Data availability

Summary statistics from all genome-wide association studies described in this paper are available from the NHGRI-EBI GWAS Catalog, accession numbers GCST90016666-GCST90016676, URL http://ftp.ebi.ac.uk/pub/databases/gwas/summary_statistics/GCST90016001-GCST90017000/GCST90016676/. All underlying data, and derived quantities, are available by application from the UK Biobank at http://www.ukbiobank.ac.uk.

The following dataset was generated:

| Author(s) | Year | Dataset title | Dataset URL | Database and Identifier |
|---|---|---|---|---|
| Cule M, Liu Y, Basty N, Whitcher B, Bell JD, Sorokin EP, van Bruggen N, Thomas EL | 2021 | Genetic architecture of 11 organ traits derived from abdominal MRI using deep learning | http://ftp.ebi.ac.uk/pub/databases/gwas/summary_statistics/GCST90016001-GCST90017000/GCST90016676/ | NHGRI-EBI GWAS Catalog, GCST900 16666-GCST90016676 |

The following previously published datasets were used:

| Author(s) | Year | Dataset title | Dataset URL | Database and Identifier |
|---|---|---|---|---|
| The Genotype-Tissue Expression (GTEx) Project | 2016 | GTEx v7 | https://storage.google-apis.com/gtex_analysis_v7/single_tissue_eqtl_data/GTEx_Analysis_v7_eQTL_all_associations.tar.gz | GTEx v7, GTEx_Analysis_v7_eQTL_all_associations |
| Abbott L, Bryant S, Churchhouse C, Ganna A, Howrigan D, Palmer D, Neale B, Walters R, Carey C, The Hail team, Anttila V, Aragam K, Baumann A, Cole J, Daly MJ, Damian R, Haas M, Hirschhorn J, Jones E, Munshi R, Rivas M, Vedantam S | 2018 | Neale lab GWAS v2 | http://www.nealelab.is/uk-biobank/ | Neale lab, GWAS v2 |
| Carvalho-Silva D, Pierleoni A, Pignatelli M, Ong CK, Fumis L, Karamanis N, Carmona M, Faulconbridge A, Hercules A, McAuley E, Miranda A, Peat G, Spitzer M, Barrett J, Hulcoop DG, Papa E, Koscielny G, Dunham I | 2018 | OpenTargets Genetics | https://genetics-docs.opentargets.org/data-access/data-download | OpenTargets Genetics, 190505 |
| Zhou W, Nielsen JB, Fritsche LG, Dey R, Gabrielsen ME, Wolford BN, LeFaive J, VandeHaar P, Gagliano SA, Gifford A, Bastarache LA, Wei WQ, Denny JC, Lin M, Hveem K, Kang HM, Abecasis GR, Willer CJ, Lee S | 2018 | Data from: Efficiently controlling for case-control imbalance and sample relatedness in large-scale genetic association studies | ftp://share.sph.umich.edu/UKBB_SAIGE_HRC/ | SAIGE UK Biobank GWAS, UKBB_SAIGE_HRC |

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

# Appendix 1

## Image preprocessing
### Dixon pipeline

The Dixon sequence involved six overlapping series that were acquired using a common set of parameters: TR = 6.67 ms, TE = 2.39/4.77 ms, FA = 10° and bandwidth = 440 Hz. The first series, over the neck, consisted of 64 slices, voxel size 2.232 × 2.232×3.0 mm and 224 × 168 matrix; series two to four (covering the chest, abdomen and pelvis) were acquired during 17 s expiration breath holds with 44 slices, voxel size 2.232 × 2.232×4.5 mm and 224 × 174 matrix; series five, covering the upper thighs, consisted of 72 slices, voxel size 2.232 × 2.232×3.5 mm and 224 × 162 matrix; series six, covering the lower thighs and knees, consisted of 64 slices, voxel size 2.232 × 2.232×4 mm and 224 × 156 matrix.

The six separate series associated with the two-point Dixon acquisition were positioned automatically after the initial location was selected by the radiographer (*Littlejohns et al., 2020*). Reconstruction of the fat and water channels from the two-point Dixon acquisition was performed on the scanner console. Four sets of DICOM files were generated for each of the six series in the neck-to-knee Dixon protocol: in-phase, opposed-phase, fat and water.

Bias-field correction (*Tustison et al., 2010*) was performed on the in-phase volume and the resulting bias field applied to the other channels (opposed-phase, fat, water) for each series. The series were resampled to a single dimension and resolution to facilitate merging the six series into a single three-dimensional volume (size = [224, 174, 370], voxel = 2.232 × 2.232 x 3.0 mm). To reduce the effect of signal loss when blending the series, we identified the fixed set of slices that form an overlap (inferior-superior direction) between adjacent series and applied a nonlinear function to blend the signal intensities on these regions of overlap. Slices in the interior of the volume were heavily weighted and slices near the boundary were suppressed. We repeated the bias-field correction on the blended in-phase volume and applied the estimated bias field to the other channels.

Fat-water swaps are a common issue in the reconstruction of Dixon acquisitions, where the fat and water labels attributed to the reconstructed images are reversed for all voxels in the acquired data series or cluster of voxels associated with separate anatomical structures (e.g., legs or arms). Once corrected, the fat and water channels are consistent. We used a convolutional neural network (CNN) model to detect swaps, with six individual models trained for each of the six acquired series. Only fat-water swaps that involved the entire series or the left-right halves in the final two series were considered. Partial fat-water swaps (e.g., the top of the liver) will be considered in future work. Each model used a sequential architecture with six layers that assigned a label to each of the series when given a central 2D slice from the series. Each convolution block ($C_n$) was made up of $n$ convolutions that were 3 × 3 spatial filters applied with stride of length two, followed by a leaky rectified linear unit (ReLU) activation with slope 0.2 and batch normalization. The final layer had stride of length three and a sigmoid activation for binary classification of the input as either water or fat. The number of convolution filters was doubled in each layer down the network as follows: $C_{64}$ - $C_{128}$ - $C_{256}$ - $C_{512}$ - $C_{1024}$ - $C_1$. The two models covering the bottom two series that include the legs checked the right and left half of the input image separately to accommodate for the legs being separate structures with increased likelihood of swaps. Each of the six series for 462 subjects were individually inspected to ensure no swaps occurred and used to train the models. The ten central coronal slices of each subject were selected by checking the image profile of the slice in each volume, where the largest profile was assumed to be the centre of the body. Thus, a total of 4620 images were available for training each of the networks. No additional data augmentation was performed. Each 2D slice was normalized. The model was trained with a binary cross entropy loss function using the Adam optimizer and a batch size of 100 until convergence, which was between 150 and 200 epochs depending on the series. The models were validated on a separate set of 615 subjects, resulting in 4920 individual swap detection operations performed as every set of Dixon data is subject to eight classification tests. The validation, via visual inspection of all the series and the swap detection results, revealed only two instances of the second series (the chest) and one instance of the fifth series (one of the two legs) were mislabelled out of the total 4920 checks performed. A single false positive, in the second series, was observed.

Anomaly detection of the final reconstructed volumes was performed to identify potential data issues such as image artifacts, positioning errors or missing series. This was achieved checking the dimensions of the final reconstructed volume and edge detection performed on the binary body mask. To generate the body mask, we applied multiscale adaptive thresholding to the flattened in-phase signal intensities, keeping only the largest connected component, then performed a binary closing operation. The presence of sharp edges in the body mask highlighted discontinuities in the data and was used as an indicator of data inconsistencies. We used Canny edge detection on a central coronal slice and a sagittal slice of the body mask containing both background and subject labels. In a normal subject, edge detection should not highlight anything other than the vertical contour of the body from neck to knee. Presence of discontinuities or horizontal features in the body mask were indicators of anomalies. Clusters of voxels in the edge image corresponding to horizontal edges exceeding a threshold 10 voxels in the sagittal and coronal slice, or 25 in either slice, triggered the anomaly detection. Those values were selected based on results of 1000 subjects. Field of view errors in positioning the subject were identified if the head or chin were partly or fully visible, or if the total volume did not match the standard $224 \times 174 \times 370$ dimension of the correctly assembled Dixon acquisition. Signal dropout artifacts were caused by metal objects such as knee or tooth implants and identified when discontinuities appeared inside the body mask.

## 3D pancreas pipeline

A high-resolution T1w acquisition sequence for determining pancreas volume was acquired under a single expiration breath hold with TR = 3.11 ms, TE = 1.15 ms, FA = 10°, bandwidth = 650 Hz, voxel size $1.1875 \times 1.1875 \times 1.6$ mm and $320 \times 260$ matrix. Two versions were provided, with and without normalization, from the scanner. Bias-field correction was performed to reduce signal inhomogeneities in the normalized volume. No additional preprocessing was applied to the high-resolution 3D T1w pancreas volumes.

## Multiecho pipeline (Gradient Echo and IDEAL)

Two types of acquisitions were performed to quantify fat in the liver and pancreas:

1. A single-slice gradient echo acquisition sequence, for both the liver and pancreas, was acquired using the common set of parameters: TR = 27 ms, TE = 2.38/4.76/7.15/9.53/11.91/14.29/16.67/19.06/21.44/23.82 ms, FA = 20°, bandwidth = 710 Hz, voxel size $2.5 \times 2.5 \times 6.0$ mm and $160 \times 160$ matrix. This acquisition was stopped for the liver after the first 10,000 subjects (approximately) and replaced by the IDEAL sequence, but was continued for the pancreas for all subjects.
2. A single-slice IDEAL sequence (*Reeder et al., 2005*) for the liver used the following parameters: TR = 14 ms, TE = 1.2/3.2/5.2/7.2/9.2/11.2 ms, FA = 5°, bandwidth = 1565 Hz, voxel size $1.719 \times 1.719 \times 10.0$ mm and $256 \times 232$ matrix.

We applied bias-field correction to each echo time separately to facilitate 2D segmentation. Software (https://github.com/marcsous/pdff) available from Dr Mark Bydder (*Bydder, 2020a*), specifically the PRESCO (Phase Regularized Estimation using Smoothing and Constrained Optimization) algorithm (*Bydder et al., 2020b*), was used to simultaneously estimate the proton density fat fraction (PDFF, referred to as fat in results) and transverse relaxivity (R2*) values voxelwise from the single-slice gradient echo (GRE) and IDEAL acquisitions. Essentially, a multi-peak spectrum was constructed from the echo times in the acquisition protocol and used to perform nonlinear least squares under multiple regularization constraints that extends the IDEAL (Iterative Decomposition of Water and Fat with Echo Asymmetry and Least-Squares Estimation) algorithm (*Reeder et al., 2005*; *Yu et al., 2008*).

For consistency with previous studies (*McKay et al., 2018*; *Wood et al., 2005*), we convert R2* into iron concentration (mg/g) using the formula: iron concentration = 0.202 + 0.0254 x R2*.

Liver iron concentrations were not adjusted for the potential effects of hepatic cellular pathologies (*Li et al., 2018*) but we would expect it to be minimal given the relatively low level of hepatocellular clinical diagnosis in the UKBB cohort.

To minimise error and confounding effects, we applied one voxel erosion to the 2D mask prior to summarising fat and iron content. If the final size was <1% of the organ's 3D volume, or <20 voxels, we excluded the mask from analysis.

To account for systematic differences between the IDEAL and GRE acquisitions, we used the acquisitions of 1487 subjects that both had GRE and IDEAL acquisitions to fit a linear model relating these two measurements. If both acquisitions were available, we used the IDEAL measurement. For those with only GRE, we used the following formulae:

$PDFF_{IDEAL} = 1.09 + 0.763 * PDFF_{GRE}$

$Iron_{IDEAL} = 0.196 + 0.855 * Iron_{GRE}$

Segmentation of organs, for volume assessment, from Dixon data

We re-purposed an updated 3D iteration of the U-net architecture (*Ronneberger et al., 2015*) based on label-free segmentation from 3D microscopy (*Ounkomol et al., 2018*). In order to produce sensible segmentations for QC purposes on minimal data, we made the following choices. Training data is intrinsically scarce, and performance can always be improved with additional data. We pursued a multi-task approach (*Zhang and Yang, 2021*) so as to improve data efficiency. The supervision loss consists of binary heads as opposed to multi-class classification because compartments can overlap spatially. We annotated multiple compartments and organs on the same individuals. Although not intrinsically novel, we are the first to scale this application to a very large UKBB imaging cohort. All weights and pipelines and data augmentation details are available to download (https://github.com/calico/ukbb-mri-sseg). This is the first time that segmentations for multiple major organs and compartments have been published on the UKBB dataset. Comparisons across datasets are also difficult because evaluation would be confounded by the specifics of how individuals are chosen, the conventions of annotation, and specifics of data acquisition or processing.

Our implementation of U-net had 72 channels on the outside, and we capped the maximum number of channels in deeper layers of the network to 1152. We used concatenation on skip connections, and convolution-transposes when upsampling. A heavily-engineered system was used to stream large datasets efficiently and perform data augmentation on demand. To address computational bottlenecks, we encoded the 3D multichannel images as urolled PNGs inside TFrecords. We relied on TensorFlow best practices to parallelise and streamline random batching during training. Data augmentation was performed on the fly on the GPU, and not pre-computed. We used a batch size of six, and some customized engineering was needed to accommodate very large tensors and total GPU memory use.

Input voxels were encoded into five channels: fat, water, in-phase, out-of-phase, and body mask. The body mask indicated whether a given voxel was inside the body The neural network branched into a different logit head for supervision on each organ. Supervision included the sum of Dice coefficient (*Milletari et al., 2016*) and binary cross-entropy across all organs.

Inspection of validation loss curves indicated that use of batch normalization and data augmentation provided sufficient regularization. During training, the model utilised 80,000 96 × 96×96 patches as subsequently described, and the Adam optimizer learning rate was reduced from 1e-5 to 1e-7 following a quadratic decay. During inference, we used Otsu thresholding (*Otsu, 1979*) to decode a binary decision for each voxel as to whether it was part of each given organ or not.

## Data augmentation

Data augmentation included a 3D deformation to locally transform 3D data smoothly as a whole, rather than by slice. We iteratively batched a small number of individual voxels, assigned random Gaussian values and convolved noise with random width Gaussian filters. The summed result was treated as a noise vector and added to the raw image dynamically. We also used a smooth elastic warp to augment the data. This augmentation assigned a different smooth 3D optical flow offset to each voxel in any spatial direction, which was effective since it could locally subsume a heterogeneous combination of commonly used spatial distortions. The same warping function was applied to training masks to ensure that supervision was consistent with input data.

Each final voxel obtained its value from a location offset by an optical flow vector sampled from a Gaussian process. To preserve visual details, voxels that were close together were sampled with strongly correlated optical flow offsets, while pairs further away were less correlated. To reduce the computational load in the optical flow sampling process, we cropped the image to a 174 × 174 ×

174 window and placed a 4 × 4 × 4 lattice of equispaced points centered inside it. These 64 lattice points had fixed relative spatial positions. Based on pairwise distances, we created a (4 × 4 × 4)-by-(4 × 4 × 4) covariance matrix to describe how correlated distortions should be in the warping. We applied a Gaussian kernel with a width of 24 voxels. These 3 × 64 values were multiplied by a random scaling chosen uniformly in [0, 4], treated as optical flow values and applied to the image in the distortion along three spatial directions for each of the 64 lattice points. Next, we extrapolated optical flow values to each underlying voxel position with a polyharmonic spline, and applied the warp by resampling the image at each voxel with its own floating point offsets in 3D. From the center of the warped and resampled image, we cropped a 96 × 96 × 96 patch and used this as training data. When interpolating supervision segmentation masks, we converted the masks to floating-point probabilities and applied clipping heuristics after the warp and resampling to ensure that probabilities were valid. Finally, we obtained volume measurements by thresholding the model output, removing disconnected structures, and multiplying the number of mask voxels by the image resolution.

Quality control consisted of iterations of visual inspection of extreme volumes for each distinct organ/structure, as well as spot checks of hundreds of random subjects. The training data was regularly enriched to include problematic cases. We repeated this procedure and retrained the model until results did not display outliers for extreme subjects nor any of the random spot checks. Performance metrics are available in *Supplementary file 1*.

## Segmentation of the liver, for fat and iron content assessment, from single-slice data

We applied a standard 2D U-net to segment the IDEAL and GRE liver data, training one model for each of the two liver acquisitions. We split 507 annotations of the IDEAL acquisition into a training set of 456 training images and 51 validation images. Similarly, we split 373 annotations of the GRE acquisition into 335 training images and 38 validation images. The unprocessed image data consisted of complex numbers in six channels in IDEAL and 10 in GRE, resulting in input shapes of (256, 232, 18) for IDEAL and (160, 160, 30) for GRE. We encoded the complex number as a triplet: magnitude, sine and cosine of the angle. We applied mild data augmentation in the form of small rotations, translations, zoom, shears, and flips. We used the Adam optimizer on 100 steps with batch size 32 for each of the following learning rates in the schedule: [1e-4, 1e-5, 1e-5, 1e-6, 1e-7]. To ensure high specificity at the cost of recall during inference (and thus ensure that our derived values do not include non-liver tissue), we used Otsu to propose a threshold based on the voxelwise prediction probabilities and adjusted the threshold to further ablate the 25% of the foreground.

