## [Decision Letter]

**Acceptance summary:**

The authors advance the understanding of abdominal organ related diseases by utilizing MRI scans, genetic information, and clinically defined trait resources from the large UK Biobank using advanced statistical methods. Ample discussion and comparison of how their results relate to known findings from existing literature is given.

**Decision letter after peer review:**

Thank you for submitting your article "Genetic architecture of 11 organ traits derived from abdominal MRI using deep learning" for consideration by *eLife*. Your article has been reviewed by 2 peer reviewers, and the evaluation has been overseen by a Reviewing Editor and Matthias Barton as the Senior Editor. The following individual involved in review of your submission has agreed to reveal their identity: Constantinos Parisinos (Reviewer #2).

The reviewers have discussed their reviews with one another, and the Reviewing Editor has drafted this to help you prepare a revised submission. Details are provided below and include recommendations relating to the introduction, readability and discussion to further improve the paper for the readers.

While you are very welcome to submit a revision, there is an alternative you may want to consider. It is possible this work would be more powerful as multiple papers. Say the following three papers, (1) How you built the IDPs, (2) IDP vs. clinically defined phenotype analyses, (3) IDP vs. genetic analyses. Or maybe (1) How you built the IDPs and the IDP vs. clinically defined phenotype analyses, (2) IDP vs. genetic analyses. That might help you get your main points across more clearly, as the key takeaways are difficult right now, given all the results.

Essential revisions:

1) Introduction

Consider starting the introduction differently, rather than with a description of the UKB project. What the problem with big data is and why the development of deep learning models for image analysis is essential for this day and age. An introduction based on the way the argument is presented in the abstract would set the story better.

2) This is a general readability comment.

There are a lot of results and methods to digest here. The Discussion section did not fully synthesize nor address all the results given. Recommend that either the Discussion section needs to be extended, or you should be more discriminatory in what is included in the Results section.

3) Methods

We would also recommend shortening some of the methods, but giving full details of them in the appendix. This would allow the reader to get a broad view of what is happening for the methods, but not get bogged down in all the details that are included right now. For example, giving a broad overview of how the IDPs were created in the Methods section in the text, but moving all the nitty gritty details to the appendix.

4) Results

A lot of interesting associations were found. Would there be any way to summarize the ones that may be clinically significant in the near future?

5) Since the exome analysis did not find any significant results, you could say that you considered an exome analysis in your methods/Results section, but move all the details and results for it to the appendix.

6) "MRI has become the gold standard for clinical research"

Too generic, MRI not the golden standard for many things, please rephrase something along the lines of "MRI is a safe, non-invasive method that can be used to accurately measure multiple phenotypes including…"

7) Was the calculated kinship matrix used in the GWAS the same as the one used in the exome-wide association study? How was the kinship matrix calculated for the GWAS (software, set of SNPs used, regions excluded, etc.)? You say you only included participants recorded as "Caucasian" for GWAS. Did you check this using PCA?

8) Did you use a set of LD pruned SNPs to calculate genomic control from the GWAS results? More attention should be given to understanding/justifying why you might have these inflated lambda values from your GWAS results. I also find it curious that many of your phenotypes have exactly the same lambda values. What do the QQplots of the GWAS p-values (separately by phenotype, using an LD pruned set of SNPs) look like (put these in supplemental)?

9) For the ICD code and raw data trait regression analyses, more detail should be given about how you ensured model stability. Having only 20 cases for your response variable in logistic regression will not give you stable results with the number of covariates you've included. What did the case/control ratio look like in your significant findings? Were a lot of your significant findings from models that had only a small number of cases? There was no mention of how many cases were needed in the "Other traits" logistic models. Did you use the "at least 20 cases" rule there too? I would expect that some of your clinical phenotypes were very skewed or zero inflated. Did you check/account for this?

10) What are the levels for the covariate Ethnicity used in your clinical trait regression analyses? Investigating how ethnicity was related to any of your clinical traits could lead to interesting discussion, e.g., were any clinical trait associations driven by the non-white subsample. This could allow for some discussion of differing disease burden through a diversity lens, even though your genetic cohort was white British.

11) Table 1 should be in the Results section Table 1 comments- The pancreas volume X % Female cell has % in it, while the others do not. Why does the 38,881 not match the 38,683 stated in the above paragraph? Are the entries for age, BMI, and height giving mean (SD)?

12) The last part of the Figure 1 legend is for E, F, and G, not just E.

13) A comment for this sentence, "Interestingly, pancreas volume was associated more strongly with Type 1 diabetes (T1D) (p=4.9e21, beta=-0.77), than T2D (p=1.1e-17, beta=-0.27), while pancreatic fat showed a small association with T2D (beta=0.181, p=1.16e-07) and not with T1D (p=0.241)." The p-value for Type 1 diabetes should be e-21, not e21. Also, are the first two p-values actually meaningfully different from each other?

14) Figure 2 – top right figure; some labelling is unclear.

15) It would be interesting to denote what is a novel association and what is a previously existing association in the Figure 3C Manhattan plot. Figure 3 – manhattan plot peaks are not labelled (e.g. with lead snp/gene) – this may be more useful

16) Could you tie in your heritability estimates for the 11 IDPs into the discussion about age-related impacts on organ function (Discussion, paragraph 2). You mention "probably reflecting genetic and environmental exposures," do your heritability estimates align with this?

17) More detail should be given about what you mean by "raw data" in the Other traits methods section. Is that text notes from the clinician?

18) There are some x-axis readability issues for your Figure S1.B plots.

19) Can you grey out the non-significant dots in Figures S8-10 (like you did for Figures S3-7)?

*Reviewer #1:*

This manuscript aims to better understand the underlying mechanisms and drivers of abdominal organ related complex diseases. The authors utilize the large amount of magnetic resonance imaging (MRI), genetic, and clinically defined trait information available in the UK Biobank to better understand complex diseases related to the liver, pancreas, kidneys, spleen, and lungs. The authors take advantage of the previously underutilized MRI scans available in UK Biobank specific to abdominal organs. Their consideration of multiple data types within UK Biobank cohort is timely given the current demand for rigorous analyses of the rapidly growing electronic health record and biobank databases.

Since most other researchers using the MRI UK Biobank data have studied cardiac and brain related traits, this manuscript gives insights to the previously understudied images of abdominal organs. The authors' detailed description of how they produced the image derived phenotypes (IDP) used in these analyses, using deep learning and the MRI scans, will be valuable to any researcher interested in using medical imaging data. The authors are the first to extend the procedure used to characterize IDPs from MRI scans to the large UK Biobank cohort. Researchers can apply to access these IDPs or the authors' code for creating them, which will progress the use of medical imaging data in future analyses in the scientific community.

The methods performed here utilize many different types of data available within the UK Biobank. Care was taken to extract clinically defined phenotypes from the data records using ICD codes and raw data fields, although some details and choices for the models used to relate these clinically defined traits to the IDP traits is lacking. The authors' approach of scanning the different data types in a pairwise fashion (IDP related to clinical phenotypes, IDP related to genetic information) is not cutting edge, as it does not integrate these multiple distinct data types together during analyses, but ultimately meets their goals of gaining a better understanding of significant associations with the complex IDPs.

The genetic analyses were conducted using only individuals of white British ancestry due to the small sample sizes of non-white populations in the UK Biobank cohort. Thus, the results from the genetic analyses may not be applicable to non-white populations. Additionally, since the clinically defined traits vs. IDP analyses were conducted using all samples, discussion relating these results to the genetic vs. IDP results is difficult, as the potential of the clinically defined trait vs. IDP associations being driven by the non-white subsample was not investigated. These issues simply serve as a reminder for the need to value diversity in the genetics community, as they could have been avoided with better recruitment efforts by genetic cohorts.

The authors support their conclusions using both their results and previously cited literature, although it is sometimes difficult to synthesize their conclusions because of the large number of results presented and analyses performed. While the authors did not have an external data set to replicate their findings, good discussion is provided to give context and support their findings using previous studies and existing literature. The analyses and given results achieve their goals of better understanding abdominal organ related diseases and of better using the MRI scans available in the UK Biobank.

*Reviewer #2:*

This is an important and very well written paper where the authors have developed deep learning algorithms to automate extractions of certain measures/phenotypes from MRI scans. They show that these measurements are associated with health and disease, and can be used with genetic data to gain new insights into biology.

Strengths

Strengths include a large sample size from an unselected cohort, with genetics, blood tests and clinical outcomes available to investigate associations.

Weaknesses

There is no validation cohort for the genetic analysis.

GWAS on White British Ancestry only.

The cohort in UKB who have undergone MR imaging is slightly healthier than the overall cohort due to some of the exclusion criteria, this should be commented on.

Although recruitment into the UK Biobank study finished in

2010, scanning began in 2014. The median follow-up period from scanning is 2.5 years, limiting power to evaluate the prognostic value of IDPs, or to evaluate whether they are a cause or consequence of the disease state.

Impact

The work is likely to significantly impact big data/ imaging research, since it is now possible to automate phenotype extraction using deep learning pipelines, compared to the much more labour intensive method of manual extraction.

---

## [Author Response]

Essential revisions:1) IntroductionConsider starting the introduction differently, rather than with a description of the UKB project. What the problem with big data is and why the development of deep learning models for image analysis is essential for this day and age. An introduction based on the way the argument is presented in the abstract would set the story better.

Thank you for this feedback. We have reframed the Introduction to introduce on the one hand, the role of MRI imaging in understanding the basis of disease, and on the other the recent explosion of biobank-scale research. We agree that this better contextualizes this work which brings together these two disparate fields.

2) This is a general readability comment.There are a lot of results and methods to digest here. The Discussion section did not fully synthesize nor address all the results given. Recommend that either the Discussion section needs to be extended, or you should be more discriminatory in what is included in the Results section.

While mindful of the length, we have expanded the discussion to:

1. Better emphasize to the deep learning innovations required to execute this work;

2. Place the genetics results in the context of previous studies.

3) MethodsWe would also recommend shortening some of the methods, but giving full details of them in the appendix. This would allow the reader to get a broad view of what is happening for the methods, but not get bogged down in all the details that are included right now. For example, giving a broad overview of how the IDPs were created in the Methods section in the text, but moving all the nitty gritty details to the appendix.

Thank you for your comments, we agree that reducing the detail within the paper would improve readability particularly for a broad audience. We have therefore added an Appendix (Appendix 1) with the full details of the methods, and substantially abbreviated the main body of the text.

4) ResultsA lot of interesting associations were found. Would there be any way to summarize the ones that may be clinically significant in the near future?

We agree with this point, with so many interesting associations, determining which have the most meaning is important. We have expanded the last paragraph of the Discussion highlighting what we perceive to be the key clinically relevant findings of the study and their potential importance for personalised medicine. We have also expanded part of the discussion to emphasize relevance for particular disease areas where this type of imaging has to date not been routinely in the clinic.

5) Since the exome analysis did not find any significant results, you could say that you considered an exome analysis in your methods/Results section, but move all the details and results for it to the appendix.

Thank you for this recommendation. while some of the exome analysis has been suggestive of interesting results, as yet nothing has demonstrated significance. We have therefore as suggested moved the details relating to the exome analysis to the Supplemental sections.

6) "MRI has become the gold standard for clinical research"Too generic, MRI not the golden standard for many things, please rephrase something along the lines of "MRI is a safe, non-invasive method that can be used to accurately measure multiple phenotypes including…"

We thank the reviewers for their comments and agree this was an overgeneralization on our part. We have now clarified our wording in the introduction of the manuscript to reflect this.

7) Was the calculated kinship matrix used in the GWAS the same as the one used in the exome-wide association study? How was the kinship matrix calculated for the GWAS (software, set of SNPs used, regions excluded, etc.)? You say you only included participants recorded as "Caucasian" for GWAS. Did you check this using PCA?

We have added additional details on the inclusion criteria, identification of Caucasian participants, and the calculation of the kinship matrix to the section “Genome-wide association study” (p. 27).

8) Did you use a set of LD pruned SNPs to calculate genomic control from the GWAS results? More attention should be given to understanding/justifying why you might have these inflated lambda values from your GWAS results. I also find it curious that many of your phenotypes have exactly the same lambda values. What do the QQplots of the GWAS p-values (separately by phenotype, using an LD pruned set of SNPs) look like (put these in supplemental)?

We have replaced the unpruned genomic control estimates from the original text/tables with an estimate from the same set of LD pruned, genotyped SNPs described above (calculated using BOLT-LMM). The different phenotypes still have very similar lambda values; we have added more significant figures to the updated ST4. We have added QQ plots (Figure 3—figure supplement 6).

9) For the ICD code and raw data trait regression analyses, more detail should be given about how you ensured model stability. Having only 20 cases for your response variable in logistic regression will not give you stable results with the number of covariates you've included. What did the case/control ratio look like in your significant findings? Were a lot of your significant findings from models that had only a small number of cases? There was no mention of how many cases were needed in the "Other traits" logistic models. Did you use the "at least 20 cases" rule there too? I would expect that some of your clinical phenotypes were very skewed or zero inflated. Did you check/account for this?

The number of cases/controls is given in Supplementary File 1d; we have added this information to Supplementary File 1e, and also logged when a logistic model was used in the ‘model’ column. While we have not conducted any formal statistical test, we did not observe any particular enrichment in traits with very small numbers of cases and the main findings were typically based on >100 cases.

10) What are the levels for the covariate Ethnicity used in your clinical trait regression analyses? Investigating how ethnicity was related to any of your clinical traits could lead to interesting discussion, e.g., were any clinical trait associations driven by the non-white subsample. This could allow for some discussion of differing disease burden through a diversity lens, even though your genetic cohort was white British.

We agree with the reviewer that ethnicity is an important covariate in this analysis and have provided more detail regarding this factor. We have added clarification that we used self-reported ethnicity from the UK Biobank. We agree that this is an interesting potential avenue for future research, especially given the observed heterogeneity in anthropometric and metabolic traits between people of different ancestries. Mindful of the length of the paper, we leave this as a potential avenue for future research.

11) Table 1 should be in the Results section Table 1 comments- The pancreas volume X % Female cell has % in it, while the others do not. Why does the 38,881 not match the 38,683 stated in the above paragraph? Are the entries for age, BMI, and height giving mean (SD)?

We have clarified the legend of Table 1, removed the extraneous % sign, and moved it to the start of the Results section. We have provided an additional table (Supplementary File 1b) which gives more context on the discrepancy between the number of scans/participants from different stages of the processing pipeline (these were not used consistently and we agree this was confusing).

12) The last part of the Figure 1 legend is for E, F, and G, not just E.

Thank you for highlighting the issue with the figure caption, we have now corrected this in the text.

13) A comment for this sentence, "Interestingly, pancreas volume was associated more strongly with Type 1 diabetes (T1D) (p=4.9e21, beta=-0.77), than T2D (p=1.1e-17, beta=-0.27), while pancreatic fat showed a small association with T2D (beta=0.181, p=1.16e-07) and not with T1D (p=0.241)." The p-value for Type 1 diabetes should be e-21, not e21. Also, are the first two p-values actually meaningfully different from each other?

We have corrected the typo. While the p-values cannot meaningfully be compared due to the differences in sample size, we have added the (approximate) 95% confidence intervals for the respective coefficients of T2D and T1D, and noted that they do not overlap.

14) Figure 2 – top right figure; some labelling is unclear.

We have abbreviated some of the overplotted text to make this plot more legible.

15) It would be interesting to denote what is a novel association and what is a previously existing association in the Figure 3C Manhattan plot. Figure 3 – manhattan plot peaks are not labelled (e.g. with lead snp/gene) – this may be more useful

We have added labels. Since these traits (except liver fat and iron) have not been studied before, we labelled all the traits.

16) Could you tie in your heritability estimates for the 11 IDPs into the discussion about age-related impacts on organ function (Discussion, paragraph 2). You mention "probably reflecting genetic and environmental exposures," do your heritability estimates align with this?

Thank you for raising this comment. The heritability estimates of the different organs have overlapping confidence intervals. However, we are not sure that this would support or refute the genetic or environmental origin of changes in organ size over time. Rather than speculate on this further (given the length of the paper), we have removed this unclear statement, and added a sentence that large-scale longitudinal datasets will be required to interrogate the genetic/environmental basis of changes in organ volume more systematically.

17) More detail should be given about what you mean by "raw data" in the Other traits methods section. Is that text notes from the clinician?

Thank you for this comment, we were referring to the original MRI images obtained from the scanner before any processing had taken place. We have clarified this by replacing the term ‘*raw data*’ with the phrase ‘*unprocessed image data*’.

18) There are some x-axis readability issues for your Figure S1.B plots.

We have improved the layout of this figure to make it more readable.

19) Can you grey out the non-significant dots in Figures S8-10 (like you did for Figures S3-7)?

This is a great suggestion; we have done this and agree that it improves readability significantly.